palaeontology/evolution/taxonomy and systematics

Phocidae, Monachinae, North Atlantic, *Callophoca*

**Author for correspondence:**
James P. Rule
e-mail: jrule.palaeo@gmail.com

# A new large-bodied Pliocene seal with unusual cutting teeth

James P. Rule[1,3], Justin W. Adams[1],
Douglass S. Rovinsky[1], David P. Hocking[2,3],
Alistair R. Evans[2,3] and Erich M. G. Fitzgerald[2,3,4,5]

[1]Department of Anatomy and Developmental Biology, and [2]School of Biological Sciences, Monash University, Melbourne, Victoria 3800, Australia
[3]Palaeontology, Museums Victoria, Melbourne, Victoria 3001, Australia
[4]National Museum of Natural History, Smithsonian Institution, Washington, DC 20560, USA
[5]Department of Life Sciences, Natural History Museum, London SW7 5BD, UK

JPR, 0000-0001-7256-2393; JWA, 0000-0002-6214-9850;
DSR, 0000-0003-4356-9523; DPH, 0000-0001-6848-1208;
ARE, 0000-0002-4078-4693

Today, monachine seals display the largest body sizes in pinnipeds. However, the evolution of larger body sizes has been difficult to assess due to the murky taxonomic status of fossil seals, including fossils referred to *Callophoca obscura*, a species thought to be present on both sides of the North Atlantic during the Neogene. Several studies have recently called into question the taxonomic validity of these fossils, especially those from the USA, as the fragmentary lectotype specimen from Belgium is of dubious diagnostic value. We find that the lectotype isolated humerus of *C. obscura* is too uninformative; thus, we designate *C. obscura* as a nomen dubium. More complete cranial and postcranial specimens from the Pliocene Yorktown Formation are described as a new taxon, *Sarcodectes magnus*. The cranial specimens display adaptations towards an enhanced ability to cut or chew prey that are unique within Phocidae, and estimates indicate *S. magnus* to be around 2.83 m in length. A parsimony phylogenetic analysis found *S. magnus* is a crown monachine. An ancestral state estimation of body length indicates that monachines did not have a remarkable size increase until the evolution of the lobodontins and miroungins.

## 1. Introduction

The Phocidae subfamily Monachinae (southern true seals) displays a high degree of morphological disparity, with species that exhibit adaptations for filter feeding [1,2], durophagy [3], longirostry [4], and both large [5] and small [6] body sizes. Recent studies have

**Figure 1.** IRSNB 1198-M203 humerus. The fragmentary lectotype of 'C. obscura' in posterior (a), anterior (b), lateral (c) and medial (d) views. Scale bar, 5 cm. Photos taken by Sébastien Beaudart.

demonstrated that extinct Monachinae displayed a greater diversity of body sizes compared with present-day phocid faunas [5,6]. However, trends in the evolution of body size within monachines have been difficult to assess in the fossil record. This is mostly due to the referral of most large monachine fossils to *C. obscura* [7]. Previous work on the phocid material of the Yorktown Formation (early Pliocene) of North Carolina has referred multiple monachine specimens to this species on the basis of large size [8]. However, these specimens have dubious association with the fragmentary lectotype (an isolated humerus) from Belgium ([7,8], figure 1), which has little to no overlap in anatomy with these more complete specimens.

As a result, *C. obscura* has become something of a wastebasket taxon, with various large fossil monachine specimens being referred to the species [9]. While progress has recently been made highlighting this problem [9,10], large isolated humeri are still often referred to this taxon. In addition, the lectotype specimen is fragmentary, and preserves few specifically diagnostic characters. This is concerning, given the recent assessment that fossil phocid humeri should be near complete in order to be diagnostic enough to form the basis of a holotype [9]. Furthermore, a morphometric analysis on phocid limb bones (including humeri) concluded that discrimination below subfamily level was poor, and isolated postcranial elements were an inadequate basis for holotypes [11].

Consequently, taxonomic revision of fossil specimens referred to *C. obscura*, plus assessment of the validity of type specimens based on isolated postcrania, is long overdue. In this study, we aim to address the status of the lectotype of *C. obscura* in the light of recent evidence [11]. In doing so, cranial and associated specimens from the Yorktown Formation will be selected for types of a distinct new species. It is on this basis that we revisit a semi-complete skull originally referred to *C. obscura* [8]. This permits new insights into the relationships of monachines, and the evolution of large body size within the group.

## 2. Material and methods

### 2.1. Institutional abbreviations

CMM, Calvert Marine Museum, Solomons, USA; IRSNB, Institut Royal des Sciences Naturelles, Brussels, Belgium; LACM, Natural History Museum of Los Angeles County, CA, USA; MNHN, Muséum national d'Histoire naturelle, Paris, France; MSNUP, Museo di Storia Naturale, Università di Pisa, Italy; MU-IMP, Monash University Integrated Morphology and Palaeontology Laboratory, Clayton, Victoria, Australia; NHMUK, Natural History Museum, London, UK; NMV, Museums Victoria, Melbourne, Victoria, Australia; SMNS, Staatliches Museum für Naturkunde Stuttgart, Stuttgart, Germany; USNM, National Museum of Natural History, Washington, DC, USA.

### 2.2. Specimens

Specimens of the new species are housed in the USNM Paleobiology collection. Fossil specimens (and SMNS 4461) were surface scanned using an Artec Space Spider structured light surface scanner, and

meshed in Artec Studio 12 (Artec 3D, Luxembourg). Specimens examined for the erection of the new fossil taxon include USNM PAL 475486 (originally referred to *C. obscura* in Koretsky & Ray [8]), USNM PAL 534034 and USNM PAL 181601. The diversity of large-sized Monachinae from the Lee Creek Mine locality and the Antwerp Basin was reassessed with specimens that have been referred to *C. obscura* (USNM V 10434, USNM PAL 186944, USNM PAL 263656, IRSNB 1116-M188, IRSNB 1156-M177, IRSNB 301) [8,9], and the holotype (USNM PAL 181419) and paratype (USNM PAL 250290) of *Auroraphoca atlantica* [9].

While most taxa were coded from first-hand observations by the authors, some taxa were coded from the literature. *Pliophoca etrusca* was coded using Berta *et al.* [10], *Nanophoca vitulinoides* was coded using Dewaele *et al.* [12], *Australophoca changorum* was partially coded using Valenzuela-Toro *et al.* [6] for the paratype material, and *Potamotherium valletoni* using Savage [13] and Tedford [14].

Rather than using the lectotype and referred material of *Leptophoca proxima*, which is composed of post-crania from the forelimb [15] and hence represents a limited sample of morphological characters, we instead opted to code a specimen from the Calvert Formation (CMM-V-2021). This specimen was referred to *Leptophoca lenis* (now a junior synonym of *Le. proxima*) in Koretsky [16] and is composed of a cranium, mandible and cervical vertebrae series. Hence, this specimen contains more morphological characters of an early phocine. Dewaele *et al.* [15] noted that due to the isolated nature of the specimen from the lectotype, and the lack of the ability to refer it to *Le. proxima*, that the specimen should not be definitively assigned to the species. Therefore, we treat it as '*Le. proxima*' CMM-V-2021 and code the specimen as an operational taxonomic unit, rather than using the character-poor lectotype material.

Similarly, the South African (Langebaanweg) fossil lobodontin *Homiphoca capensis* has had multiple isolated postcranial specimens referred to it, despite the lack of associated specimens that can be reliably referred to the cranial holotype specimen [17–19]. So far, *Ho. capensis* is the only named phocid from this site. While an initial study indicated that there may be multiple species of seal present from Langebaanweg [19], follow-up work failed to support this [20]. As a result, we have coded both crania and postcrania referred to *Ho. capensis* from Langebaanweg. This is important as the humeri from Langebaanweg possess 'classical' monachine morphology, but also possess an entepicondylar foramen, a feature traditionally considered as exclusive to Phocinae [21,22].

Additionally, we have treated *Monotherium? wymani* as has been recently suggested by Dewaele *et al.* [23], who restricted the name to the holotype, and treated the holotype as the temporals only (dismissing a partial fibula that was previously assigned to the holotype). We coded all other fossil specimens from holotypes where possible, and only used referred specimens when there was reasonable basis for referral. Specimens used for character coding and as comparative material are listed in electronic supplementary material.

Due to the limited availability of specimens with broken tympanic cavities available for study, comparisons of the middle ear cavity and ossicles will not be made. The relatively incomplete nature of most of the postcrania of USNM PAL 534034 limits meaningful comparisons of postcrania to the humerus.

## 2.3. Measurements

Linear measurements of fossil specimens were taken using digital calipers, reported to the nearest 0.01 mm. Measurements of the holotype and paratype specimens are presented in tables 1–3. More detailed measurements of the holotype skull (USNM PAL 475486) are present in Koretsky & Ray [8].

## 2.4. Terminology

Anatomical terminology follows that used in Amson & de Muizon [3] and Berta *et al.* [10]. For all other terminology not covered by these studies, we referred to Evans & de Lahunta [24].

## 2.5. Body size estimation

Various equations for estimating body length in Phocidae were used. Due to the incomplete condition of most fossil skulls of phocids, neither of the preferred stepwise or complete subsets equations (which use multiple skull metrics) for Phocidae could be used to predict body mass or body length [25]. Instead, the single variable linear regressions for regions preserved in the skull were used. Metrics for these regressions were taken for the length (LB, $R^2 = 0.72$, per cent prediction error (PPE) = 3.43) and width (WB, $R^2 = 0.86$, PPE = 3.17) of the auditory bulla, length of the tooth row (LUTR, $R^2 = 0.78$, PPE = 4.97)

**Table 1.** Measurements of USNM PAL 475486, fragmentary skull.

| measurement | value (mm) |
|---|---|
| total preserved length (rostral fragment) | 150.97 |
| right IOF height | 8.82 |
| nasal aperture length (from anterior-lateral extent of nasal to medial tuberosity of premaxilla) | 70.17 |
| nasal aperture width (from most lateral extents of medial margins of premaxilla) | 40.15 |
| right tooth-row length (anterior canine to posterior P5) | 111.08 |
| incisor alveoli row width (distance between lateralmost extents) | 26.39 |
| canine alveoli widths (distance between lateralmost extents) | 54.86 |
| width at narrowing of tooth row between P1 and P2 (distance between lateral extents of alveoli) | 47.52 |
| left P2 anterior–posterior crown length | 16.69 |
| left P2 medial–lateral crown width | 8.31 |
| left P2 crown height | 7 |
| right P3 anterior–posterior crown length | 15.42 |
| right P3 medial–lateral crown width | 7.72 |
| right P3 crown height | 7.38 |
| right canine crown + root length | 53.39 |
| right canine crown height | 18.63 |
| right nasal bone length (anterior–posterior) | 75.27 |
| basioccipital width between medial borders of jugular foramen | 66.02 |
| left zygomatic arch (anterior–posterior) length (from anterior jugal to posterior glenoid process) | 138.48 |
| bizygomatic width (measured from surface scan) | 180.7 |
| preserved orbit width (most anterior to posterior jugal length) | 69.13 |
| left glenoid fossa width (medial–lateral) | 39.21 |
| left glenoid fossa length (anterior–posterior) | 20.83 |
| left tympanic bulla width at anterior extent (medial–lateral) | 54.75 |
| left tympano-mastoid length (anterior–medial of bulla to posterior–lateral of mastoid) | 67.7 |
| right glenoid fossa width (medial–lateral) | 38.32 |
| right glenoid fossa length (anterior–posterior) | 21.05 |
| right tympanic bulla width at anterior extent (medial–lateral) | 55.87 |
| right tympano-mastoid length (anterior–medial of bulla to posterior–lateral of mastoid) | 71.58 |

**Table 2.** Measurements of USNM PAL 534034, ear region (right side) and humerus.

| measurement | value (mm) |
|---|---|
| anterior–posterior length of the tympano-mastoid region | 80.72 |
| right glenoid fossa length (ant–pos) | 22 |
| right glenoid fossa width (medial–lateral) | 32.48 |
| humerus length (proximal–distal) | 180.44 |
| distal epicondyle of humerus (medial–lateral length) | 69.56 |
| maximum preserved width of deltopectoral crest of humerus (anterior–posterior) | 65.86 |
| maximum preserved length of deltopectoral crest of humerus (proximal–distal) | 114.55 |
| medial epicondyle width of humerus (anterior–posterior) | 40.62 |

**Table 3.** Measurements of USNM PAL 181601, left ear region.

| measurement | value (mm) |
|---|---|
| preserved glenoid fossa width (medial–lateral) | 33.71 |
| preserved glenoid fossa length (anterior–posterior) | 26.51 |
| tympanic bulla width at anterior extent (medial–lateral) | 58.78 |
| tympano-mastoid length (anterior–medial of bulla to posterior–lateral of mastoid) | 77.72 |
| tympanic bulla length (anterior–posterior) | 64.03 |

and length of the postcanine tooth row (LUPC, $R^2 = 0.65$, PPE = 5.80) and width across the canines (CW, $R^2 = 0.91$, PPE = 2.98) following table 5 of Churchill *et al.* [25, p. 239]. In addition to the regressions, body length estimates based on the humerus were calculated using reported ratios [26,27] of the humerus length to total body length of *Ommatophoca rossii* and *Leptonychotes weddelli*, as used in a recent study [12]. Details regarding metrics and estimation methods used are outlined in electronic supplementary material. When multiple methods were used for estimating total body length for fossil taxa, the mean estimate was reported. This was as long as the $R^2$ and PPE values were within a similar range, so that the precision of a mean value was not biased by an estimated value with poor fit.

The above methods were implemented for taxa (*Monotherium? wymani*, *Homiphoca* sp., *A. atlantica*, *Virginiaphoca magurai*, *Hadrokirus martini* and *Pl. etrusca*) which did not have total body length estimates in Churchill *et al.* [5] and Valenzuela-Toro *et al.* [6], and are reported in electronic supplementary material. For the new taxon, metrics of the skull for USNM PAL 475486 (LUTR, LUPC) and USNM PAL 181601 (LB, WB), and the total length of the humerus of USNM PAL 534034, were used for body length estimations. These mean estimations were then used in the maximum-likelihood ancestral state estimation. Total body length records for extant taxa were also taken from King [21] to aid in comparisons and maximum-likelihood ancestral state estimation (see the section below).

Total body length estimates and values were then collated and plotted to compare the range of sizes of phocids among four assemblages: the Pisco Formation (Upper Miocene), the Yorktown Formation (Lower Pliocene), the recent western North Atlantic and the recent Southern Ocean. The Pisco Formation phocids consist of *Au. changorum*, *Acrophoca longirostris*, *Piscophoca pacifica* and *Ha. martini*. The Yorktown Formation phocids consist of the new taxon, *Homiphoca* sp., *A. atlantica*, *V. magurai* and a phocine fossil referred to 'Gryphoca similis' (USNM PAL 263625, [8]). Several fossils from the Yorktown Formation were referred to various phocine taxa in Koretsky & Ray [8]. However, it has been recently highlighted that these specimens do not represent the various taxa they were referred to [28] and so may need to be revised. Therefore, for the sake of comparison, we will simply treat these fossils as 'Phocinae'. The recent phocids of the western North Atlantic consist of *Cystophora cristata*, *Halichoerus grypus*, *Pagophilus groenlandicus*, *Phoca vitulina* and *Pusa hispida*. Other extralimital geographical records from additional arctic phocines exist, but these taxa were not included for comparison. The recent monachines of the Southern Ocean consist of *Mirounga leonina*, *Lobodon carcinophaga*, *O. rossii*, *Hydrurga leptonyx* and *L. weddellii*.

## 2.6. Phylogenetic analysis

A phylogenetic analysis was performed using TNT (Willi Hennig Society 2016), using the traditional search option, equal weighting. Max trees were set at 50 000, using Wagner trees set at 1000 replications, and a tree bisection reconnection (TBR) as the swapping algorithm. Bootstrap values were calculated using 1000 replications. When possible, 168 morphological characters were coded for each operational taxonomic unit, including 25 that are either novel or developed from the literature (listed in electronic supplementary material). All 168 characters coded were unordered. The phylogeny of recent taxa was constrained using a molecular tree [29].

All 21 recent species and 11 fossil species (*Enaliarctos emlongi*, *Allodesmus kernensis*, *Devinophoca claytoni*, 'Leptophoca. proxima,' *Au. changorum*, *Ac. longirostris*, *Ha. martini*, *Pi. pacifica*, *Ho. capensis*, *Monotherium? wymani* and new taxon) were coded directly from specimens. *Pliophoca etrusca*, *N. vitulinoides*, the *Au. changorum* paratype and *Po. valletoni* were coded from the literature [10,12–14]. *Potamotherium valletoni* was used as an outgroup taxon, due to its putative affinities as a freshwater stem-pinniped [3,30].

Branch lengths and node ages for the phylogeny were calculated using the 'strap' package for R-Studio version 1.2.1335 (R v. 3.6.0). Node ages are reported in electronic supplementary material. In addition, a maximum-likelihood ancestral state estimation was performed in R-Studio (using the package 'geiger' to import and prune trees, and 'phytools' to run the analysis) for total body length in Monachinae. This used total body length of extant taxa (including *Neomonachus tropicalis*) and estimated total body length of fossil taxa for the maximum-likelihood ancestral state estimation (electronic supplementary material). We mapped the ancestral state estimation onto the consensus tree of the equal weights parsimony analysis.

# 3. Results

**Systematic Palaeontology**

Order CARNIVORA Bowdich, 1821 [31]

Suborder PINNIPEDIA Illiger, 1811 [32]

Family PHOCIDAE Gray, 1821 [33]

Subfamily Monachinae Gray, 1869 [34]

Genus *Sarcodectes* gen. nov.

*LSID.* urn:lsid:zoobank.org:act:57FC2732-1EEA-4765-A698-F4DD9DF30E33.

*Type and only included species*. *Sarcodectes magnus* sp. nov.

*Diagnosis of genus*. The same as type and only includes species.

*Etymology.* Derived from the Greek noun 'sárx' meaning 'flesh', and the Greek noun 'dektes' meaning 'biter', in references to the dental and musculoskeletal adaptations to processing flesh.

*Sarcodectes magnus* sp. nov.

*Callophoca obscura* Van Beneden 1876 [7] (in part, Koretsky & Ray, 2008 [8, p. 109])

'*Callophoca obscura*' (Berta *et al*. 2015 [10, p. 23])

Monachinae indet. (Dewaele *et al.,* 2018 [9, p. 10])

*LSID.* urn:lsid:zoobank.org:act:EF7293A3-D821-43F8-B8FF-56CA3D86EC21.

*Etymology.* The Latin 'magnus' meaning large. This is in reference to the large body size relative to other fossil phocid taxa.

*Diagnosis.* A phocid on the basis of: the bulbous inflated tympanic bulla, nasal inserted between the frontals and a pachyosteosclerotic mastoid. A monachine due to: the presence of four instead of six upper incisors (the latter state for all phocines except *C. cristata*); the premaxilla forming the lateral border of the nasal aperture, which is partially obscured in lateral view; the carotid foramen being visible in ventral view and a mastoid that is obscured in dorsal view.

A large seal, of an adult size comparable to the extant *H. leptonyx*. Differentiated from all other phocids by the following combination of features: on the upper postcanines, the height of the main cusp is less than twice the height of the accessory cusps, with thin carinae on both the mesial and distal edge (absent in all phocids except Monachini); distinct intercuspid notch between the main cusps and immediate distal accessory cusp; the postcanine teeth are not obliquely orientated (unlike Monachini), with the cusps of each tooth orientated in the mesial–distal plane; the diastema separating the PC4 and PC5 is large relative to the other diastemas in the upper postcanine tooth row (present in both 'stem' and crown-lobodontins to the exception of *Lobodon* and *Ommatophoca*); the lateral surface of the posteroventral apex of the tympanic bulla covers the petrosal, but is not covered by the mastoid (similar to *Mirounga*); the mastoid, on the posteroventral surface, possesses an enlarged, convex protuberance that flares laterally, which is lacking in other monachines; viewed dorsally, the region of the squamosal that lies dorsal to the mastoid barely extends past the auricular opening, resulting in a lack of a post-auricular shelf (also present in *Mirounga*, *Hydrurga* and *Monotherium? wymani*); the external auditory meatus extends anteriorly as a groove onto the post-glenoid process (also present on *Erignathus*, and more subtly on *Cystophora*); possesses a large, rounded, antero-posteriorly thick medial epicondyle on the humerus, with a proximally located fossa for the origin of the pronator teres on its anterior surface; and possesses a combination of a medio-laterally thickened deltopectoral crest that is obliterated on the shaft distally (in posterior view), no fossa for the medial head of the triceps brachii, a virtually absent radial fossa, a shallow olecranon fossa and a lateral epicondylar crest that does not project posteriorly.

*Holotype.* USNM PAL 475486 (figures 2–4). Collected by C. Swindell. Fragmentary but mostly undistorted cranium consisting of complete premaxilla, semi-complete maxilla consisting of the right canine, right PC2–PC3, and left PC2; nasal, interorbital with fragmentary frontal, complete left

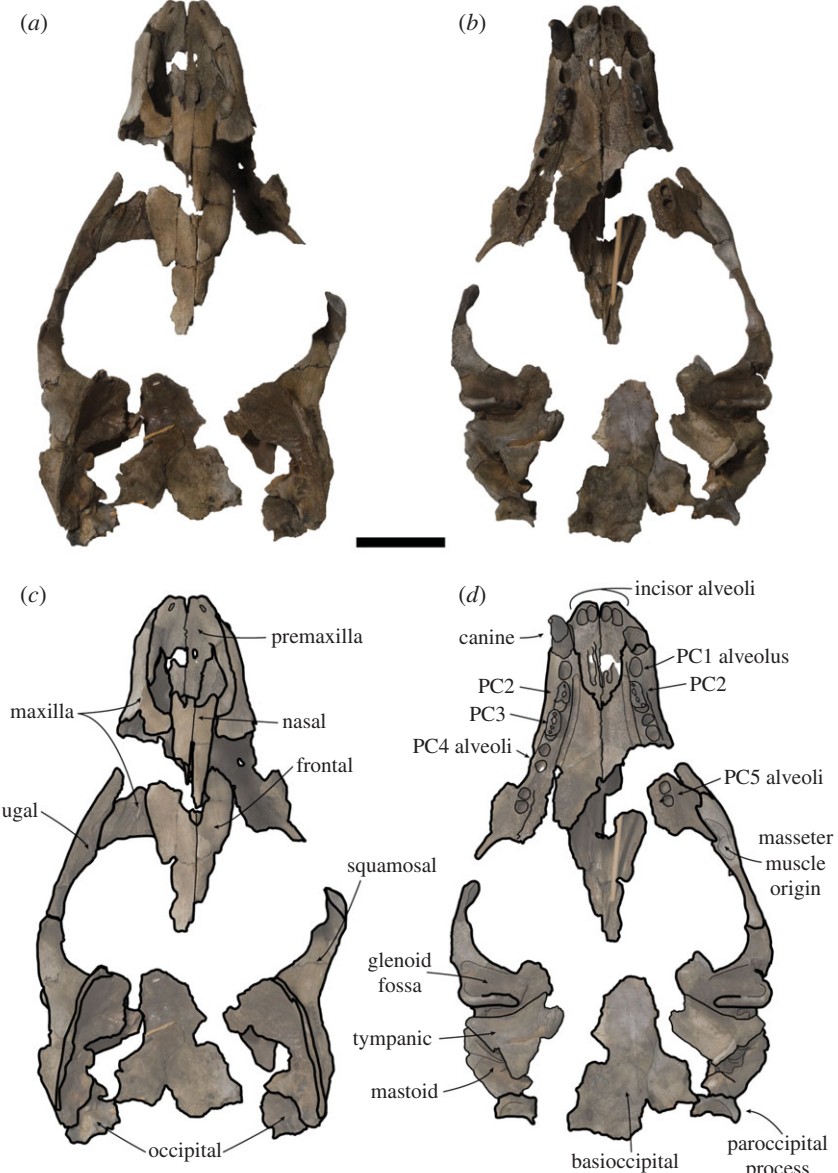

**Figure 2.** USNM PAL 475486 skull. Photos (*a,b*), and drawings (*c,d*) of holotype in dorsal (*a,c*) and ventral (*b,d*) view. Skull fragments oriented in approximate anatomical position. PC = postcanine. Scale bar, 5 cm.

zygomatic arch and incomplete right zygomatic arch; fragmentary right and left temporals, with partial basioccipital, left paroccipital process and fragment of the occipital condyle.

**Paratypes.** USNM PAL 534034 (figures 5 and 6). Collected by F.V.H. Grady. Partial skeleton consisting of fragmentary basicranium including temporal, paroccipital process, basisphenoid fragment, occipital condyle fragment, fragments of interorbital region; fragmentary cervical vertebrae; scapula fragment; semi-complete humerus; partial proximal radius and distal ulna; metacarpal and first phalanx; rib fragments; manubrium and sternebrae.

USNM PAL 181601 (figures 7 and 8). Collected by P.J. Harmatuk. Complete left temporal, including glenoid fossa.

**Type locality.** Lee Creek Mine, south side of Pamlico River, Beaufort County, near Aurora, NC, USA; USNM Locality 42246 (Lat 35 23 40 N, Long 76 47 47 W).

**Type horizon and age.** Chesapeake Group, Yorktown Formation. Dated as early Pliocene (Zanclean), according to Akers [35] and Ward & Blackwelder [36]. The age of this formation was more recently reassessed, and was constrained to 4.9–3.9 Ma [37–41].

**Comments.** The incomplete cranium USNM PAL 475486 was previously referred to *C. obscura* in a study on the Lee Creek Mine (North Carolina, USA) phocid fossils [8]. This is despite the selected

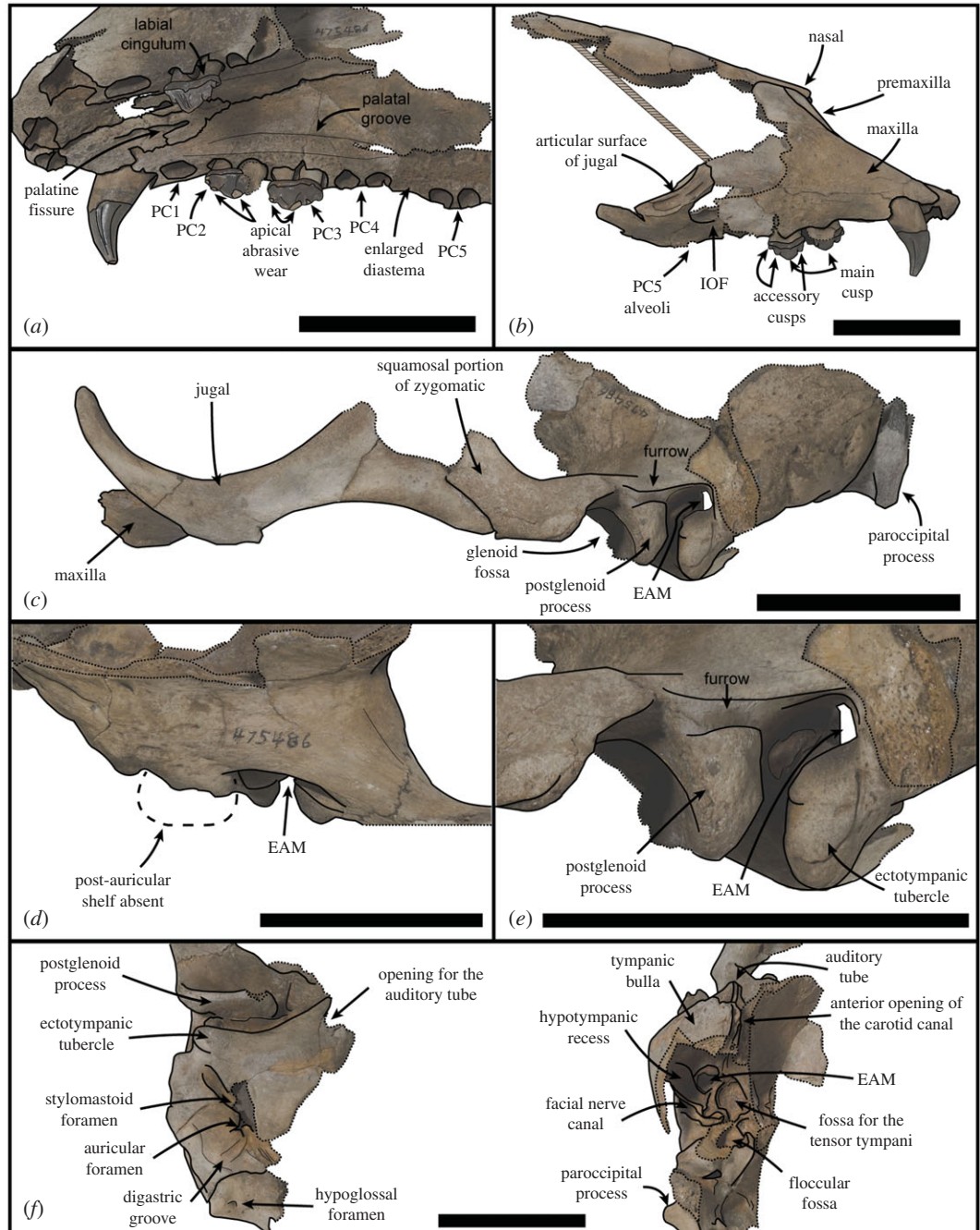

**Figure 3.** USNM PAL 475486 skull. (*a*) Oblique left ventrolateral view of rostrum. (*b*) Right lateral view of rostrum. (*c*) Lateral view of left fragment including maxilla, zygomatic arch and braincase. (*d*) Dorsal view of right squamosal. Large dashed lines indicate regular monachine post-auricular shelf. (*e*) Lateral view of left squamosal. (*f*) Right ear region, left image ventral view, right image medial view. EAM = external auditory meatus, IOF = infraorbital foramen, PC = postcanine. Small dashed lines indicate broken edges. Oblique lines represent stilt used to support specimen. Scale bars, 5 cm.

isolated lectotype humerus being non-comparable to USNM PAL 475486 (a cranium), and originating from the late Miocene to early Pliocene of the Antwerp Basin in Belgium [7,9]. The lectotype selected by Koretsky & Ray [8] was the same specimen originally described by Van Beneden (figure 1, [7]). However, Koretsky & Ray [8] referred multiple specimens (including USNM PAL 475486) from the Lee Creek Mine to *C. obscura*, based solely on large size. Initial doubt in the referral of Lee Creek material to the Antwerp Basin *C. obscura* was highlighted by Berta *et al.* [10], who suggested the taxonomy of these specimens needed to be revisited. Dewaele *et al.* [9], upon reassessing these Lee Creek specimens, concluded that all non-humeri specimens be treated as Monachinae indet. due to their being incomparable with the lectotype.

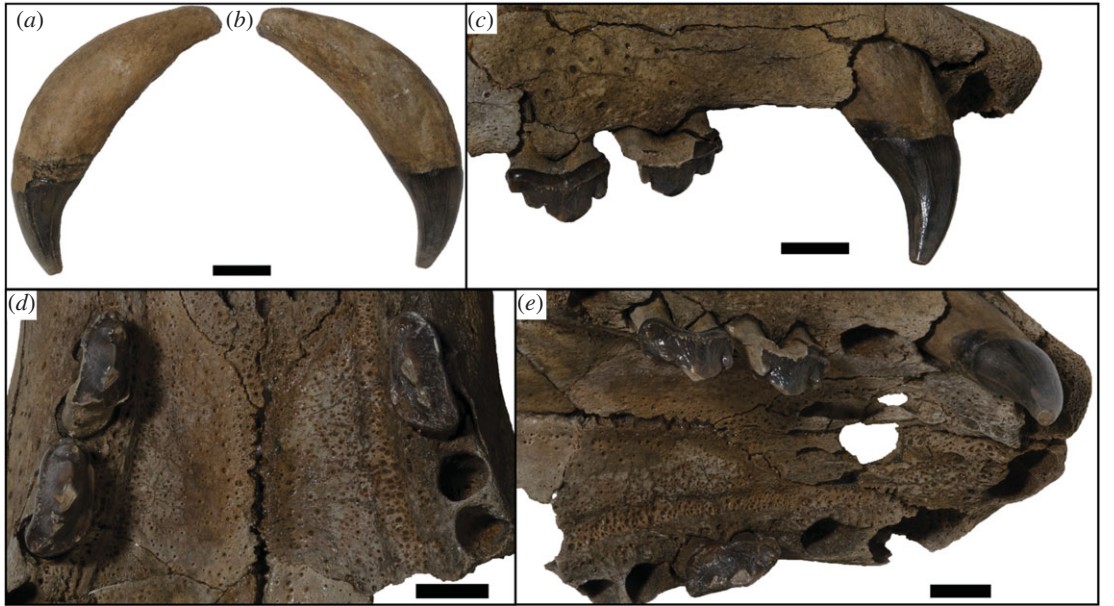

**Figure 4.** USNM PAL 475486 dentition. Canine in lingual (*a*) and labial (*b*) views. (*c*) Right lateral view of the tooth row. (*d*) Ventral (occlusal) view of the tooth rows. (*e*) Oblique posterolateral view of the tooth rows. Scale bars, 1 cm.

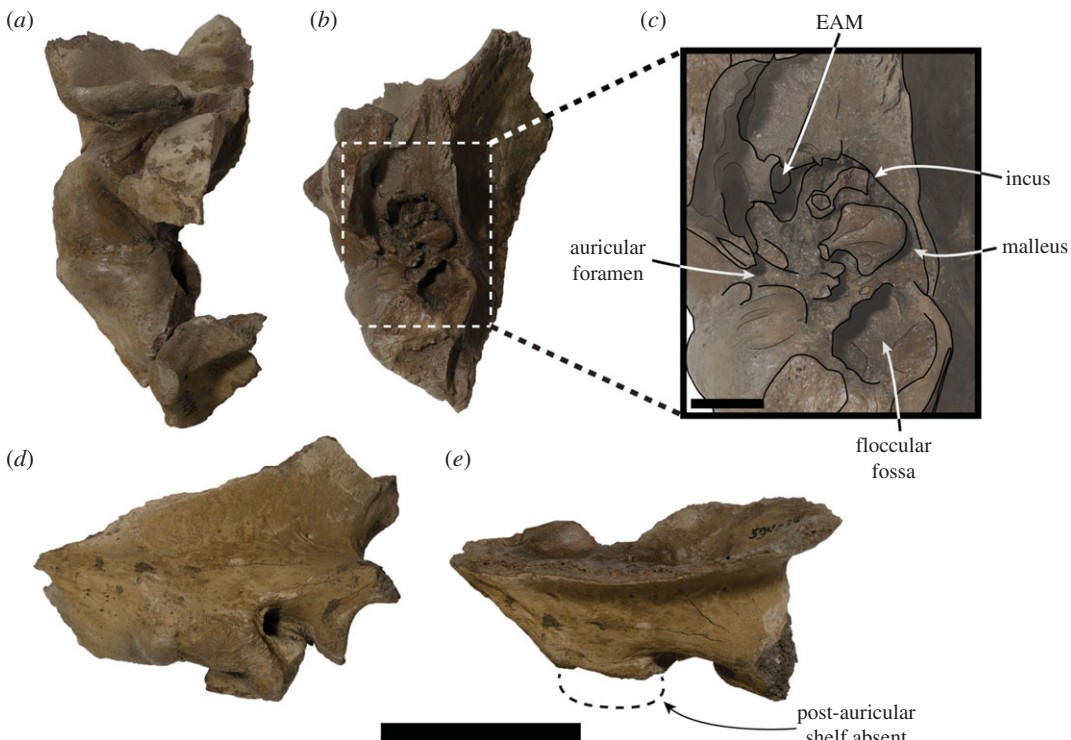

**Figure 5.** USNM PAL 534034 right squamosal. (*a*) Ventral view and (*b*) medial view. (*c*) Close-up illustrated medial view of area designated by box in (*b*). (*d*) Lateral view. (*e*) Dorsal view, large dashed lines indicate regular monachine post-auricular shelf. EAM = external auditory meatus. Scale bar,  5 cm (*a,b,d,e*), and 1 cm (*c*).

However, when assessing the condition of the lectotype, referral of even the humeri is questionable. The lectotype humerus (IRSNB 1198-M203) is fragmentary, lacking taxonomically informative features of the proximal portion, with only the anterior side of the distal end being complete enough to warrant comparison (figure 1). In essence, it completely lacks any informative characters to assign it to a specific rank, making assignment below Monachinae doubtful. Additionally, Dewaele *et al.* [9] found that humeri are only diagnostic when in a complete (or near complete) state of preservation, due to

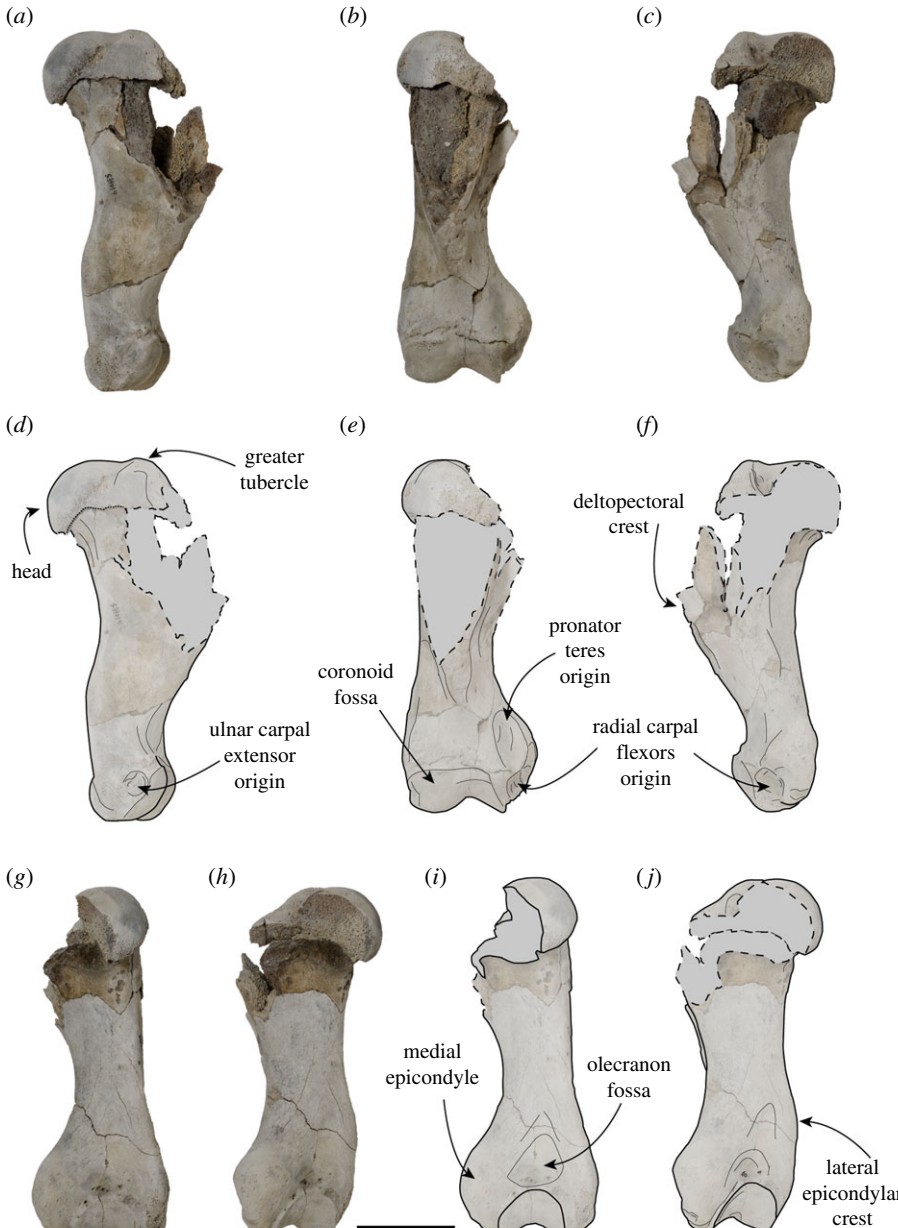

**Figure 6.** USNM PAL 534034 right humerus. Photos (*a–c,g,h*) and drawings (*d–f,i,j*) of right humerus. (*a,d*) Lateral view, (*b,e*) anterior view, (*c,f*) medial view, (*g,i*) posterior view and (*h,j*) posteromedial view. Dashed lines and grey areas in drawings indicate broken regions. Scale bar, 5 cm.

limited morphologically variable characters in humeri. However, a morphometric analysis of phocid humeri (and femora) found that there was extensive overlap in the morphospace of different species [11]. As a result, humeri could not be distinguished below subfamily level [11]. The study concluded that fossil phocid taxonomy required substantial revision, as some taxa described on the basis of isolated postcranial type specimens are potential nomina dubia.

Based on their morphometric analysis and qualitative review of humeral characters, Churchill & Uhen [11] concluded that humeri are non-diagnostic and inadequate holotypes. Consequently, *C. obscura* is a nomen dubium, as the lectotype specimen from Belgium is fragmentary and non-diagnostic. We recommend restricting the name *C. obscura* to the lectotype (IRSNB 1198-M203). *Mesotaria ambigua*, a taxon that was previously synonymized with *C. obscura* [8], and previously referred to the genus (as *Callophoca ambigua*, [42]), should also have its name restricted to the type specimen (IRSNB 1156-M177). This is because the lectotype specimen is also an isolated humerus (and thus is an uninformative type), and is currently without an anatomical diagnosis in the literature. We treat the taxa as '*M. ambigua*' as recommended by Churchill & Uhen [11], and recommend specimens

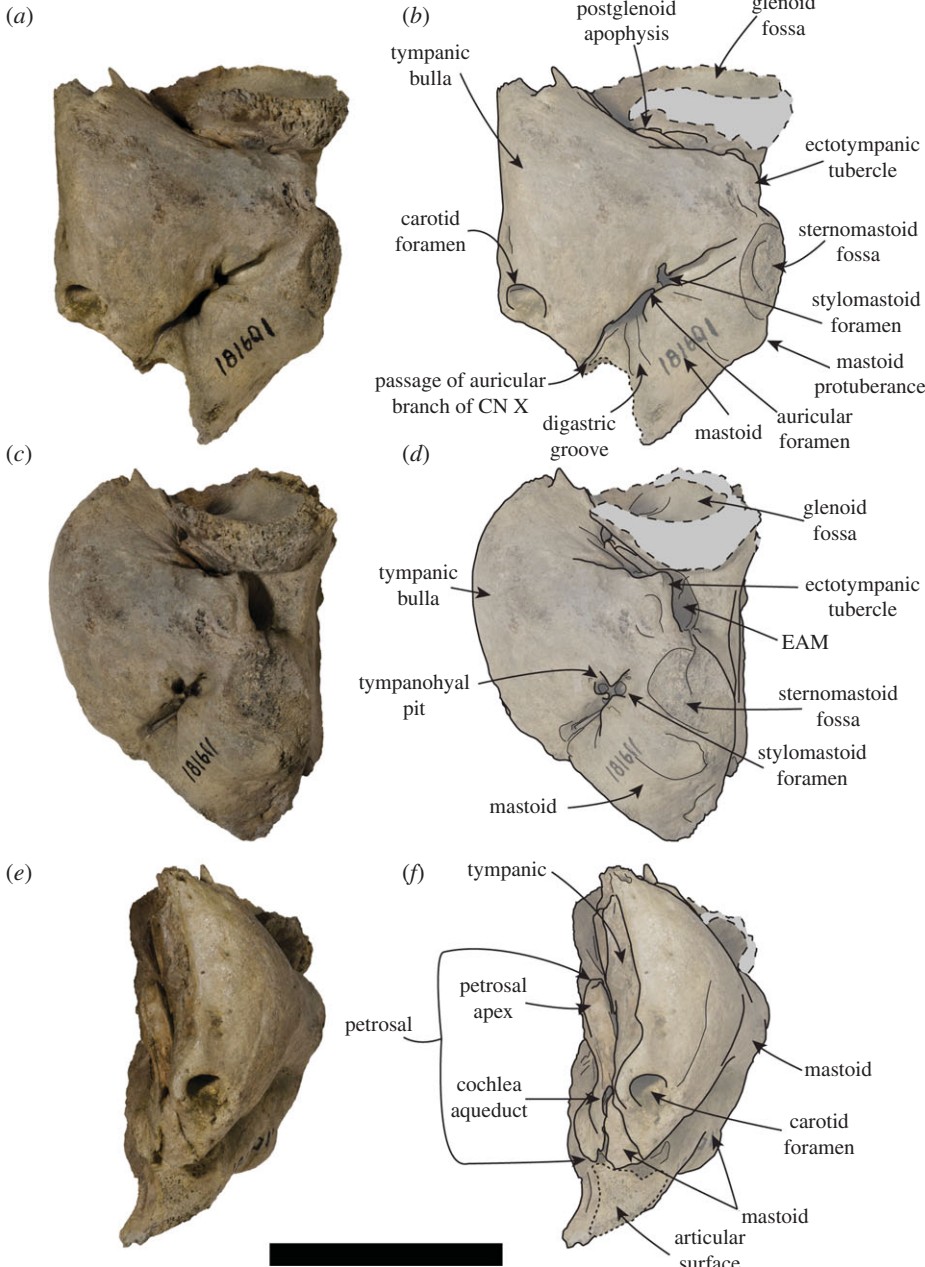

**Figure 7.** USNM PAL 181601 left ear region. Photos (*a,c,e*) and drawings (*b,d,f*) of the left ear region. (*a,b*) Ventral view, (*c,d*) ventrolateral view and (*e,f*) medial view. Large dashed lines and grey areas indicate broken surfaces. Small dashed lines indicate articular surface. CN = cranial nerve, EAM = external auditory meatus. Scale bar, 5 cm.

are not referred to this name unless there is a complete humerus that can be demonstrated beyond doubt to be morphologically identical to the lectotype.

Due to this, the incomplete cranium (USNM PAL 475486) from the Lee Creek Mine can be assigned the new binomial of *S. magnus*. The paratype specimen, which includes a cranium associated with a fragmentary humerus (USNM PAL 534034), allows this new taxon to be differentiated from other large monachines such as '*M. ambigua*' and '*A. atlantica*'.

## 3.1. Description

### 3.1.1. General

Unless stated otherwise, the description of the cranium is based on the holotype specimen USNM PAL 475486 (figure 2). The description of the ear region is mostly based on the paratype USNM PAL 181601

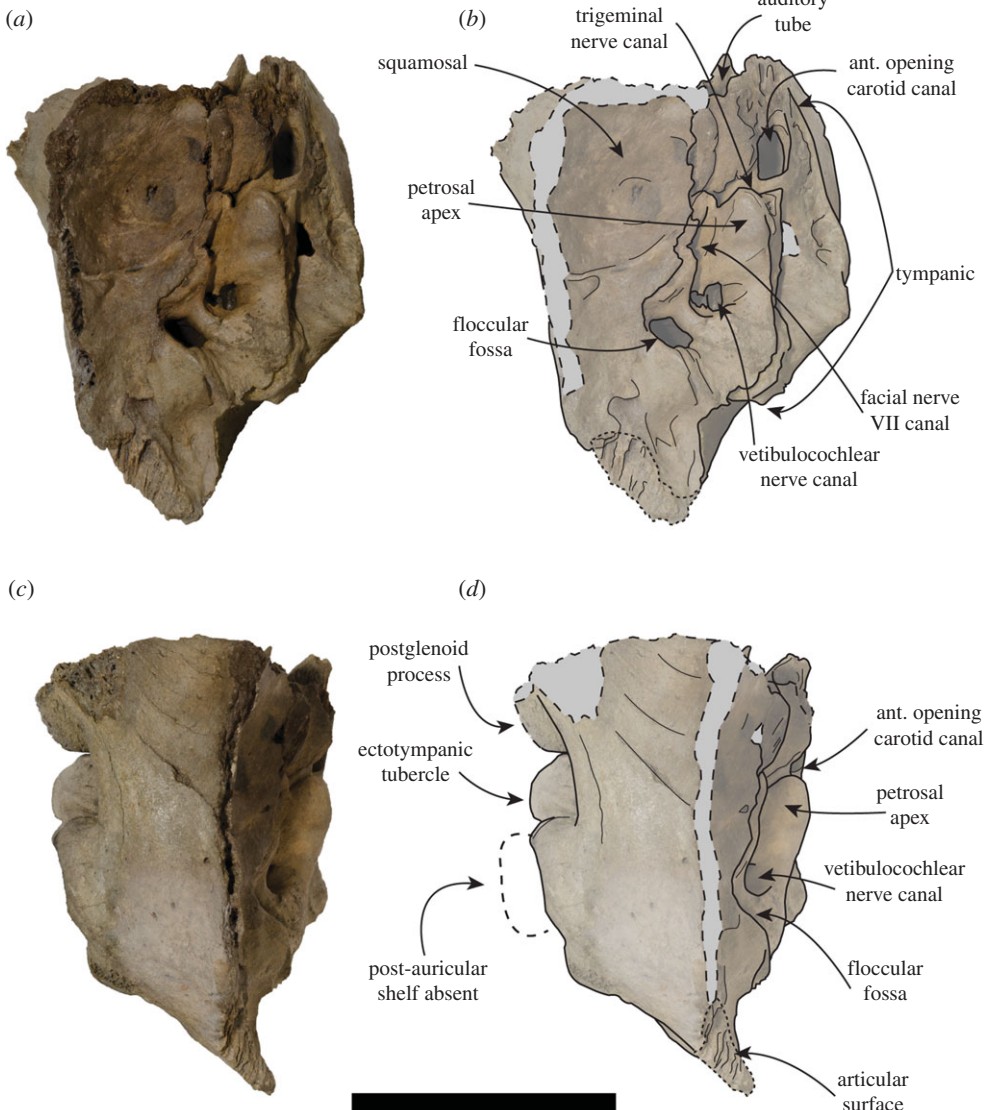

**Figure 8.** USNM PAL 181601 left ear region. Photos (*a,c*) and drawings (*b,d*) of the left ear region. (*a,b*) Medial view and (*c,d*) dorsal view. Large dashed lines with grey areas indicate broken surfaces. Small dashed lines indicate articular surface. Large dashed lines with white area indicate regular monachine post-auricular shelf. ant = anterior. Scale bar, 5 cm.

due to its relatively complete preservation (figures 7 and 8). Postcranial descriptions are based on the paratype USNM PAL 534034 (figure 6), as this is the only specimen represented by postcrania associated with cranial elements. Due to the relative incompleteness of the majority of the postcrania, only the cervical vertebrae and humerus are described in detail. All three specimens are united based on shared morphology in the temporal region (outlined in the diagnosis).

When the specimen designated as the holotype was originally referred to *C. obscura* [8], it was identified as a juvenile individual on the basis of unworn teeth and open sutures. However, the main and minor cusps clearly demonstrate some apical abrasive wear on their lingual apices (figures 3*a* and 4). In addition, the only visibly open sutures appear to be the palate and rostral region (figures 2 and 3*a*). These sutures have been shown to stay open relatively late into adulthood in phocids, sometimes never closing [43,44]. A better indicator of minimal adult age in phocid skulls is the basisphenoid–basioccipital suture [43]. As this suture appears closed in the holotype (figure 2), the specimen is at minimum a subadult, but probably represents a skeletally mature adult.

The same can be said for the paratype USNM PAL 534034, due to the apparent suture obliteration in the basicranial fragment preserved (electronic supplementary material, figure S3). USNM PAL 534034 also preserves a nuchal crest, indicating skeletal maturity (electronic supplementary material, figure S3). While the proximal end of the humerus of USNM PAL 534034 is unfused, many otherwise

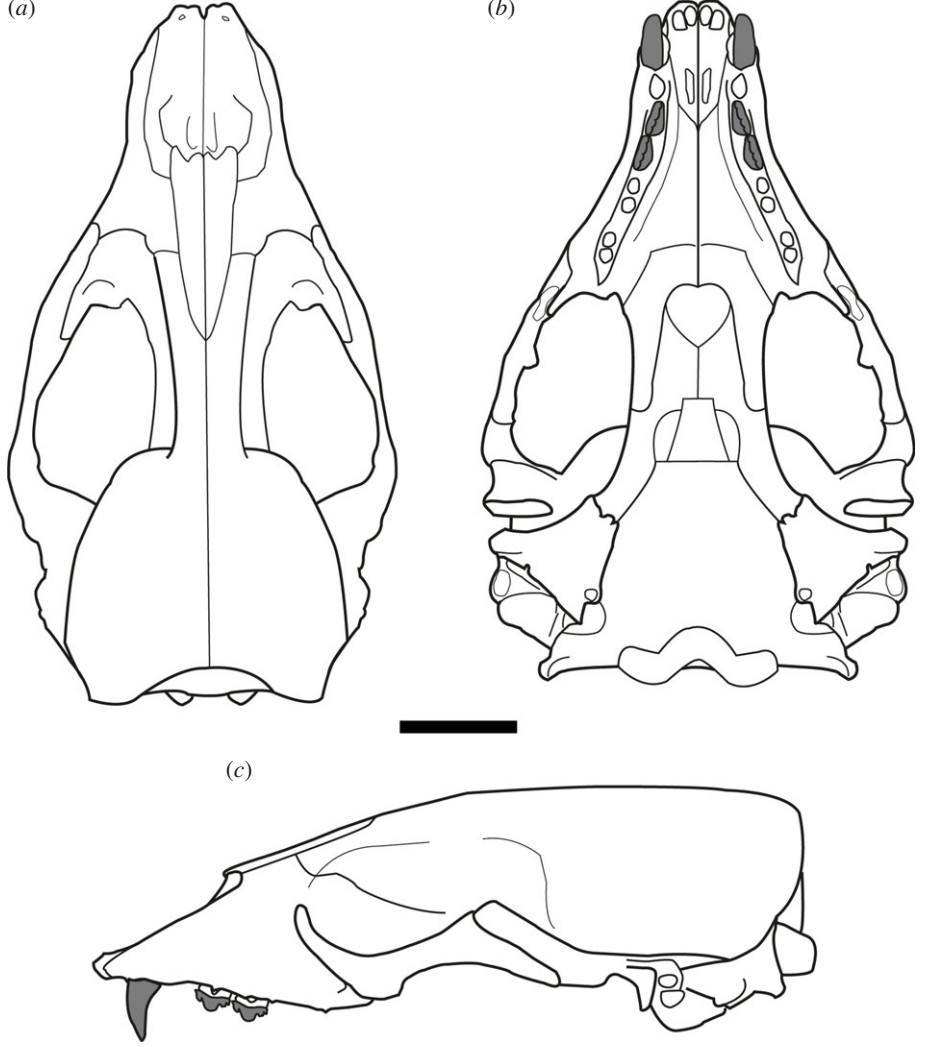

**Figure 9.** Reconstruction of *S. magnus*. Line drawing reconstruction of the cranium in dorsal (*a*), ventral (*b*) and left lateral (*c*) views. Scale bar, 5 cm.

skeletally mature phocids display apparently immature postcrania (see NHMUK 1951.8.28.16, NHMUK 1959.12.17.4). It is on this basis that the paratype (USNM PAL 534034) also probably represents a subadult at least, probably a skeletally mature adult. As the other paratype (USNM PAL 181601) is only represented by an isolated temporal, less can be definitively stated about its skeletal maturity. However, due to its size relative to the other types (tables 1–3), and the lack of the entotympanic–ectotympanic suture [45], it probably also represents a skeletally mature adult.

The cranial remains and preserved postcrania for all type specimens are robust relative to other phocids. A three-dimensional reconstruction of the skull demonstrates a dorsoventrally tall cranium ([46], figure 9). The palatal region is wide relative to the preserved basicranial region, and the cranium is broadest at the zygomatic arches.

### 3.1.2. Cranium

The anterior end of the skull preserves the rostrum, palate, jugals and the anterodorsal end of the interorbital region. The interorbital region shows that the midsagittal plane is defined by a dorsal apex on the skull, possibly representing an elongated sagittal crest (figure 2). The nasal bones are inserted into the frontal bones, the suture forming an elongated 'v' shape. The anterior end of the nasals terminates into a double arch, with the nasals protruding at the midline and on the lateral aspects. The premaxilla slopes dorsally up the medial nasal aperture, ending in a contact with the nasals. The anterior end of the premaxilla, at the tip of the rostrum, is blunt and ends in a small medial tuberosity. The nasal aperture is tall, and elongated in lateral view (figure 3*b*). The palatine fissure is preserved within the nasal cavity, and is

relatively small in dorsal view, but elongated in ventral view (figure 3*a*). In anterior view, the infraorbital foramen (IOF) is semi-ovoid and obliquely orientated. Immediately anterior to the IOF, the surface of the maxilla furrows out into an elongated, concave antorbital fossa, located above the PC3 and PC4. The PC5 is located ventral to the lateral border of the IOF (figure 3*a*).

There is a distinct portion of maxilla separating the jugal from the tooth row laterally. In lateral view (figure 3*c*), the jugal is relatively dorsoventrally thin at the anterior end, and dorsoventrally thick at the posterior end. This gives the dorsal profile of the ventral rim of the orbit a tight curvature (figure 3*c*). The orbit itself is large (approx. 7 cm, 38.24% of bizygomatic breadth, electronic supplementary material). On the ventral side of the jugal, there is an enlarged anterior surface for the origin of the masseter, extending to almost the entire ventral length of the orbit (figure 2*d*). The zygomatic process of the squamosal is shorter than the jugal, and is dorsoventrally thick. A distinct space separates the jugal from the glenoid fossa ventrally. This is also reflected dorsally on the squamosal, with the zygomatic process well separated from the braincase. A large postglenoid process results in a deep glenoid fossa. When reconstructed, the glenoid fossae appear to be orientated sub-parallel to each other along the transverse plane, although this may be heavily affected by taphonomic distortion of the zygomatic regions.

In dorsal view, the squamosal fossa lateral to the braincase is reduced (figures 3*d*, 5*e* and 8*d*). Despite this, the mastoid is completely obscured in dorsal view. In ventral view, the mastoid is pachyosteosclerotic and large, with a large protuberance posterolaterally. This results in an enlarged attachment for the digastric muscle. The mastoid is medially framed by a robust and dorsoventrally elongate paroccipital process. On the most lateral portion of the mastoid, there is a large rugosity for the sternomastoid muscle (figure 7). Immediately anterior to this rugosity is the external auditory meatus. The external auditory meatus is unobscured laterally, circular in shape, with the canal entering the braincase in a posteromedial direction. Directly anterior from the external auditory meatus, there is an extended furrow, located on the lateral surface of the postglenoid process (figure 3*e*). The ventral border of the furrow is defined by a lip on the postglenoid process. The furrow is narrowest posteriorly, being more open on the anterior end.

The inflated tympanic bulla is triangular in ventral view. The ectotympanic tubercle extends lateral to the mid-point (transversely) of the glenoid fossa, but not lateral to its lateral edge and forms the lateral–ventral floor of the external auditory meatus. In the paratype USNM PAL 181601, the tympanohyal pit is open, shallow and located on the posterolateral surface of the bulla (figure 7*d*). There is only a minor sulcus located ventral from the tympanohyal pit. The stylomastoid and auricular foramen are partially separated by the tympanic bulla and the mastoid. There is a shallow digastric groove leading to the external cochlear foramen, present on the holotype (figure 3*f*) and both paratypes (figures 5 and 7*b*). The mastoid does not contact the tympanic bulla, and the posterior end of the tympanic bulla obscures the petrosal in ventral view. The carotid foramen is visible in ventral view, with the tympanic extending into a posterior apex medially to it.

The dorsal surfaces of the tympanic and the petrosal are preserved in the paratype USNM PAL 181601. The anterior opening of the carotid canal is large and ovoid. There is a triangular flange of tympanic medial to the petrosal, which is directly dorsal to the carotid foramen. The petrosal apex is broad, rounded and dorsoventrally thickened. Directly lateral is the canal for facial nerve, which is separate from the fossa for the vestibulocochlear nerve (figure 8*b*). There is only a small remnant of the roof of the internal auditory meatus. The canal for the vestibulocochlear nerve is large, and medially located in the fossa for the vestibulocochlear nerve. A small vascular canal is located lateral to the canal for the vestibulocochlear nerve in the fossa, with the wall of bone separating them damaged (figure 8*b*). Lateral to this complex is the floccular fossa, which is large, ovoid and clearly open in dorsal view. Posterior to this is the cochlear aqueduct, which is small and slit-like (figure 7*f*). The mastoid region located medial to the cochlear aqueduct is small, but takes up a significant portion of the petrosal, and does not expand past the posteromedial border of the tympanic (figures 7*f* and 8*b*).

Both the holotype (USNM PAL 475486) and the paratype (USNM PAL 534034) provide views of the middle ear due to the breaks in the specimens in medial view. The internal opening of the external auditory meatus is relatively narrow and ovoid, opening up into a large tympanic cavity, with a pyriform and laterally deep hypotympanic recess located ventrally (figure 3*f*). Posterodorsal to the external auditory meatus is a large fossa for the tensor tympani muscle. Anterior to this fossa is a slight groove on the interior anterolateral wall of the bulla, which tapers into a narrow auditory tube. The walls of the tympanic bulla are thick, with the thickest cross-section being at the most ventral apex of the bulla. Posteromedial to the external auditory meatus and medial to the fossa for the tensor tympani muscle is the fossa for the middle ear ossicles (epitympanic recess, figure 5*c*). The malleus has a circular head with a broad contact for the tympanic membrane, with an elongated body leading to the contact with the incus. The incus is much smaller, with a relatively elongate body, and takes up

a smaller portion of the fossa. In medial cross-section of the petrosal, and located posterior to the fossa for the tensor tympani muscle, is the facial nerve canal (leading from the auricular foramen) (figures 3*f* and 5*c*). Directly posterior to this is the floccular fossa, which is enlarged internally (relative to its opening), and is reniform in cross-section (figure 5*c*).

The dental formula for *S. magnus* is 2I/1C/4P/1M. Due to the homodont nature of phocid premolars and molars, we here refer to them simply as postcanines (PC1–PC5). There are four upper incisor alveoli, with the distal incisors being only slightly larger than mesial ones. The incisor row is slightly curved. The canine is preserved on the right side only, with the canine strongly recurved posteriorly (figure 4*a,b*). The root is enlarged relative to the crown. Apical abrasive wear at the tip of the canine has resulted in a blunting of the apex. The postcanine tooth rows diverge posteriorly from each other (30° from the midline), with the palate narrowest between the PC1s (figure 2*b*). The PC1 alveolus is located directly posterior of the canine alveolus with little diastema. The PC2 and PC3 both have a dominant main cusp with one smaller mesial and two smaller distal accessory cusps, with the most distal accessory cusps the smallest and forming part of the distal cingulum (figures 3*b* and 4*c,d*). The main cusp has carinae on both the mesial and distal side, and the first distal accessory cusp has carinae posteriorly (figure 4*d*). There is apical abrasive wear present on the apices of the main and accessory cusps (with the exception of the distal-most accessory cusp), resulting in blunting of the apex of these cusps (figures 3*a* and 4*d,e*). This wear is most prominent on the lingual side of the apices (figure 3*a*).

Distinct notches are present between the main cusp and the mesial and distal cusps (figures 3*b* and 4*c*). There is a prominent cingulum present. The distal–lingual surface of the crown projects lingually to form a small occlusal shelf (figure 4*d*). The crowns, main and accessory cusps, and carinae of the preserved right PC2 and PC3 are aligned in the same orientation mesiodistally (figures 2 and 4*d*). The preserved postcanines are of a similar size, with the alveoli of PC2–PC4 being of a similar length. The PC5 alveoli are slightly smaller and separated from PC4 by an enlarged diastema (figure 3*a*). The palatal groove is relatively large and medio-laterally thick, extending from the palatal foramen at the posterior PC4 to the PC1 (figure 3*a*). The concave furrow from the groove results in the postcanine tooth row being separated ventrally from the palate by a distinct shelf.

### 3.1.3. Postcranial skeleton

The cervical vertebra that is most complete is a post-axis (C3–C7) vertebrae. Both the anterior and posterior epiphyses are missing. Only the body and the left transverse process are preserved. The body is dorsoventrally flattened. The transverse foramen is large relative to the transverse process. The transverse process is posteriorly directed, with the tubercle and lamina poorly isolated from one another. The lamina is antero-posteriorly wide and flattened.

The humerus of USNM PAL 534034 is complete distally, with the proximal section missing the lesser tubercle, the medial section of the head and the lateral side of the deltopectoral crest (figure 6). The proximal epiphysis is unfused. In posterior view, the body distal to the head is lacking a developed fossa for the medial head of the triceps. The head is broad antero-posteriorly, with the greater tubercle being level with the head proximally. The deltopectoral crest is slightly medially directed, and from the section preserved appears broad and tapers to the distal end of the humerus uninterrupted. In anterior view, the deltopectoral crest is medio-laterally thick, with the medial and distal edges tapering distally.

The humerus is well developed proximally, when compared with the distal end. The lateral epicondylar crest is slight in profile and does not project posteriorly, but is robust. There is a small, rounded fossa present for the origin of the ulnar carpal extensor (figure 6*d*). The medial epicondyle is well developed, is robust and broad in posterior view, and antero-posteriorly thickened in medial view. The medial epicondyle has a proximodistally elongate and curved medial profile, and projects medially relative to the coronoid. In anterior view, the medial epicondyle is concave, with a deep fossa for the origin of the radial carpal flexors present (figure 6*e,f*). There is a large area proximal to this fossa for the origin of the pronator teres (figure 6*e*). Both the olecranon fossa and radial fossa are shallow. The coronoid fossa is broad antero-posteriorly, with the lateral section larger than the medial section.

## 3.2. Comparisons

### 3.2.1. General

For the majority of this section, we will make comparisons only with monachine taxa; however, an exception will be made for several distinct craniodental features. The main and distal accessory cusps

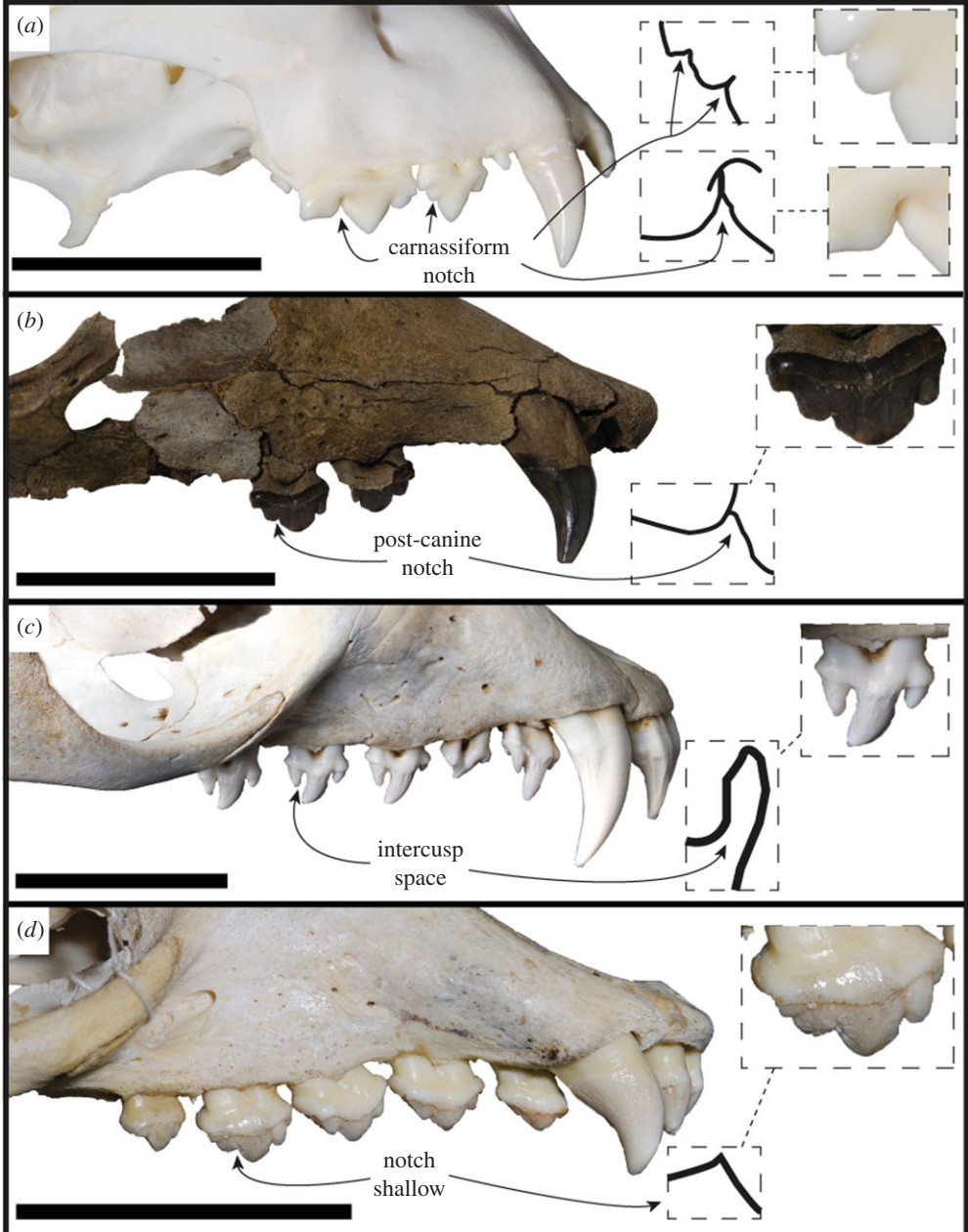

**Figure 10.** Intercuspid notch condition. Lateral view of tooth rows of *Acinonyx jubatus* MU-IMP 501 (*a*), *S. magnus* USNM PAL 475486 (*b*), *H. leptonyx* NMV C31561 (*c*) and *Neomonachus schauinslandi* USNM 243849 (photo horizontally flipped) (*d*). Drawings represent the profile of the intercusp space of the indicated tooth. Scale bars, 5 cm.

of the *S. magnus* postcanines closely abut, and the confluence of their respective carinae form a strong intercusp notch (*sensu* [47]; figure 10). These intercusp notches match a score of 4 out of a possible 5 on the Hartstone-Rose [47] metric. This differs from the condition found in all other phocids (except *Noriphoca gaudini* and *Ac. longirostris*), where a reduction or lack of this structure is due to either: small or absent accessory cusps; an enlarged separation and spacing of cusps (as in some phocins and lobodontins, e.g. *H. leptonyx*; figure 10*c*); or an accessory cusp that is not isolated but does not closely abut the main cusp, resulting in a broader angle (in labial view) between cusps (e.g. *Neomonachus schauinslandi*; figure 10*d*). The intercusp notch is present in the carnassiform dentition of the stem-pinniped *Enaliarctos* [48], and so may represent a plesiomorphic condition for pinnipeds.

A continuous cutting surface in line with the functional axis of the tooth row is present in the preserved postcanine dentition (right PC2–PC3) of *S. magnus* (figure 11). Additionally, the diastema between PC3 and PC4 is similarly sized to the PC2–PC3 diastema, and the PC4 alveolus is in line with the PC3. In all likelihood, the PC4 may have formed a continuation of the PC2–PC3 carinal line.

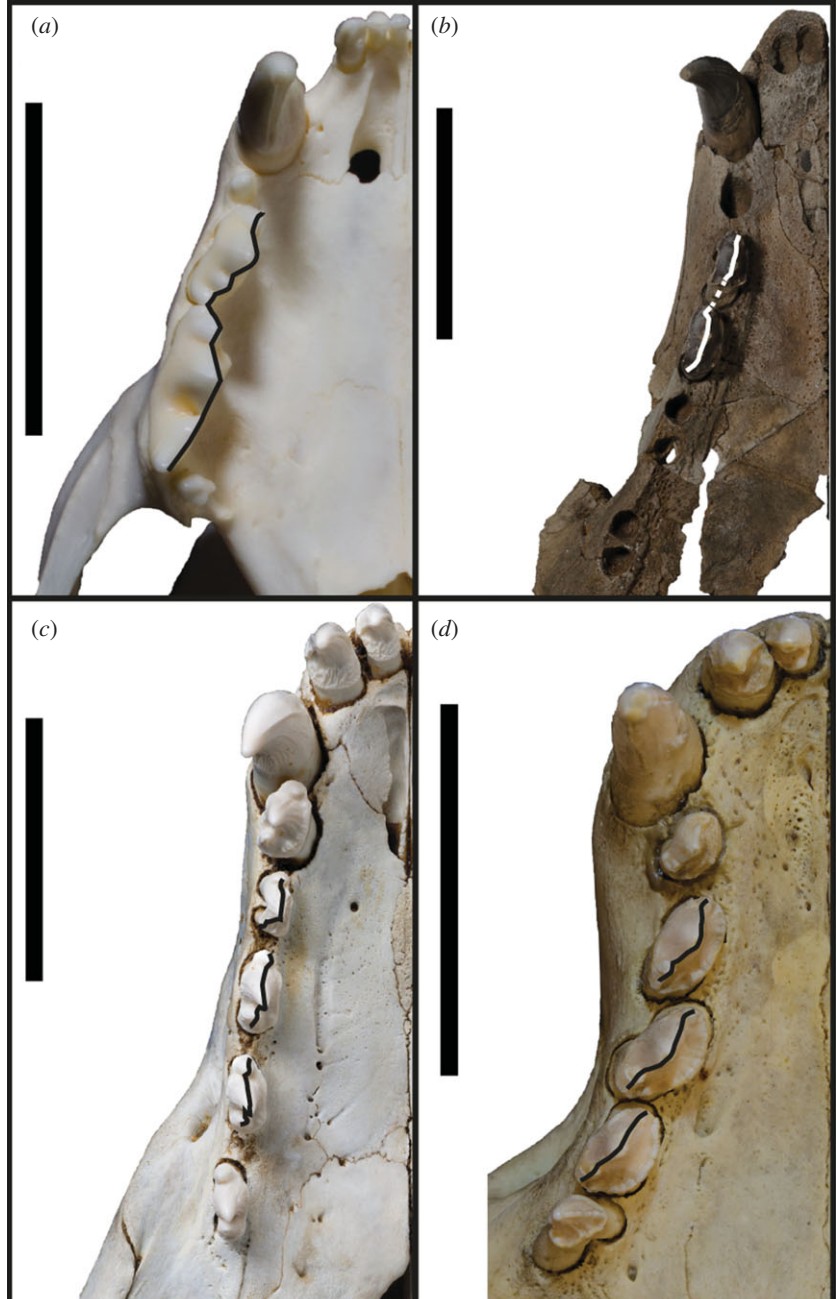

**Figure 11.** Profile of postcanines. Ventral view of the right tooth row of *A. jubatus* MU-IMP 501 (*a*), *S. magnus* USNM PAL 475486 (*b*), *H. leptonyx* NMV C31561 (*c*) and *M. monachus* NHMUK 1894.7.27.2 (*d*). Drawn-on lines indicate the profile of edges of tooth or carinae, with dotted line in *b* extrapolated from the corresponding left P2. Scale bars, 5 cm.

This differs from the usual condition of diastemata separating the postcanines from each other (figure 11*c*) or the postcanines forming an oblique orientation (figure 11*d*). An adaptation commonly present in terrestrial carnivorans adapted to produce higher bite forces is the presence of an enlarged origin for the masseter muscle on the ventral surface of the zygomatic arch (figure 12). This attachment is enlarged relative to the usual condition in phocids, and interrupts the continuous medial surface of the jugal (figure 12*a,b*). In most phocids where a visible muscle attachment is present (such as *H. leptonyx* and *Monachus monachus*), this origin is relatively small and does not interrupt the profile of the arch. For the carnivorous adaptations described above, more specific monachine comparisons are made in the following section detailing the cranium.

The anterior extension of the external auditory meatus onto the postglenoid process (figures 3*e* and 13*a*) appears to only be shared by a few taxa. It is most distinctive in *Erignathus* (figure 13*b*), forming a deep furrow on the postglenoid process. However, in juvenile *Erignathus* specimens (figure 13*c*), this

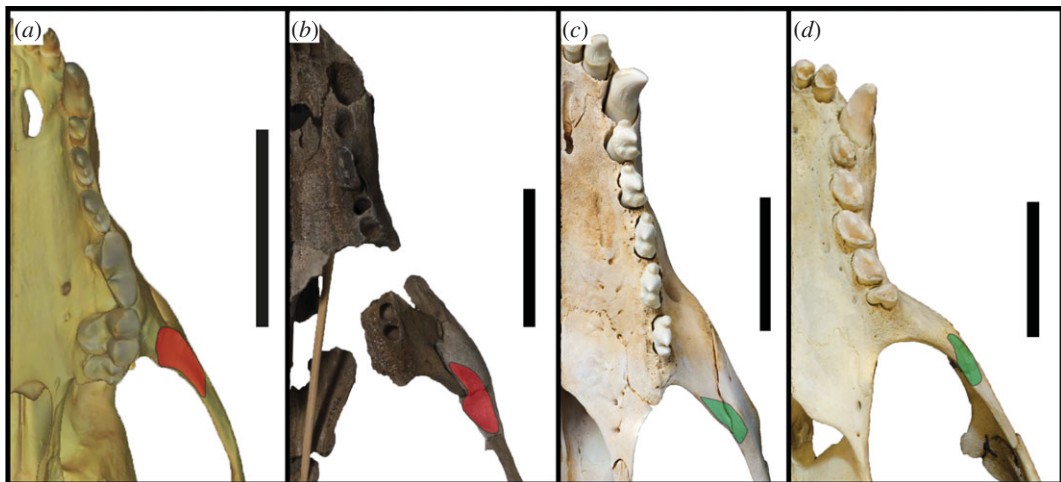

**Figure 12.** Origin of masseter muscle. Ventral view of the left tooth row and zygomatic arch of *Lycaon pictus* SMNS 4461 (surface scan) (*a*), *S. magnus* USNM PAL 475486 (*b*), *H. leptonyx* NMV C27418 (*c*) and *M. monachus* NHMUK 1894.7.27.2 (*d*). Red patch indicates enlarged masseter origin that enlarges and interrupts the zygomatic arch profile. Green patch indicates reduced origin of masseter. Scale bars, 5 cm.

structure is more poorly developed, indicating that it may vary with ontogeny. There additionally appears to be a slight variation of this feature on *Cystophora* and *Mirounga*; *Cystophora* only has a short and narrow furrow present, while *Mirounga* has a wide and shallow furrow. When considering if this character is an ancestral state, it should be noted that it is lacking on the stem-phocid *Devinophoca*, as well as outgroup taxa in Otariidae and Odobenidae. While the external auditory meatus postglenoid groove in *Erignathus* appears to be most similar to *S. magnus*, there is a deeper transition to the opening of the external auditory meatus in *S. magnus* (figure 13*a*). By contrast, the transition from the groove to the opening of the external auditory meatus in *Erignathus* is quite abrupt on the lateral surface of the skull (figure 13).

### 3.2.2. Cranium

The nasal aperture of *S. magnus* is antero-posteriorly elongated (figure 3*b*), a feature present in most monachines with the exception of *M. monachus*, *L. weddelli* and *O. rossii*. The nutrient foramen on the premaxilla of the holotype is quite large (figure 2*c*), although this feature appears to be variable within species when compared with modern specimens. A contacting nasal and premaxilla is a feature shared with the Monachini and *L. weddelli*. The IOF of *S. magnus* is of a relatively large size, a feature it shares with all monachines except *O. rossii* (which has a smaller IOF). The IOF is located dorsal to the PC5 (figure 3*b*), which *S. magnus* shares with all monachines except crown-lobodontins, where the IOF opens anterior to PC5.

Like the majority of Monachinae, the distal incisor alveoli are larger than the mesial incisor alveoli. The incisor alveoli are in transverse alignment with one another, with only a slight curve as is present in *Ho. capensis*. The canine of *S. magnus* is similar in size and shape to those of *Neomonachus* and *Lo. carcinophaga*, and is not as robust as the canines of *Mirounga*, *L. weddelli* and *H. leptonyx*. The palate is broader at the posterior end relative to the narrower anterior end, a feature shared with the monachins, fossil lobodontins (except *Ac. longirostris*), *No. gaudini*, *Lo. carcinophaga* and *O. rossii*. However, the divergence between the tooth rows in *S. magnus* is much greater than in these taxa (except *O. rossii*), resulting in a much wider posterior palate. In lateral view, the palate is oriented from anterodorsal to posteroventral, a condition similar to (but not as exaggerated as) the one present in *Pi. pacifica*, *Ha. martini* and *Ho. capensis*. The palate is flattened at the anterior end in the transverse plane, similar to most monachines, but different to the arched palates present in *Mirounga* spp., *H. leptonyx*, *M. monachus*, *Pi. pacifica* and *Ha. martini*.

The postcanine dentition of *S. magnus* differs markedly from most named phocid taxa. The unworn postcanines dentition of phocines (such as *Phoca*) sometimes displays elongated accessory cusps distinctly separate from the main cusp. However, this differs markedly from the notch morphology in *S. magnus* (figure 3*a,b*). The carinae present on mesial and distal edges of the main and accessory

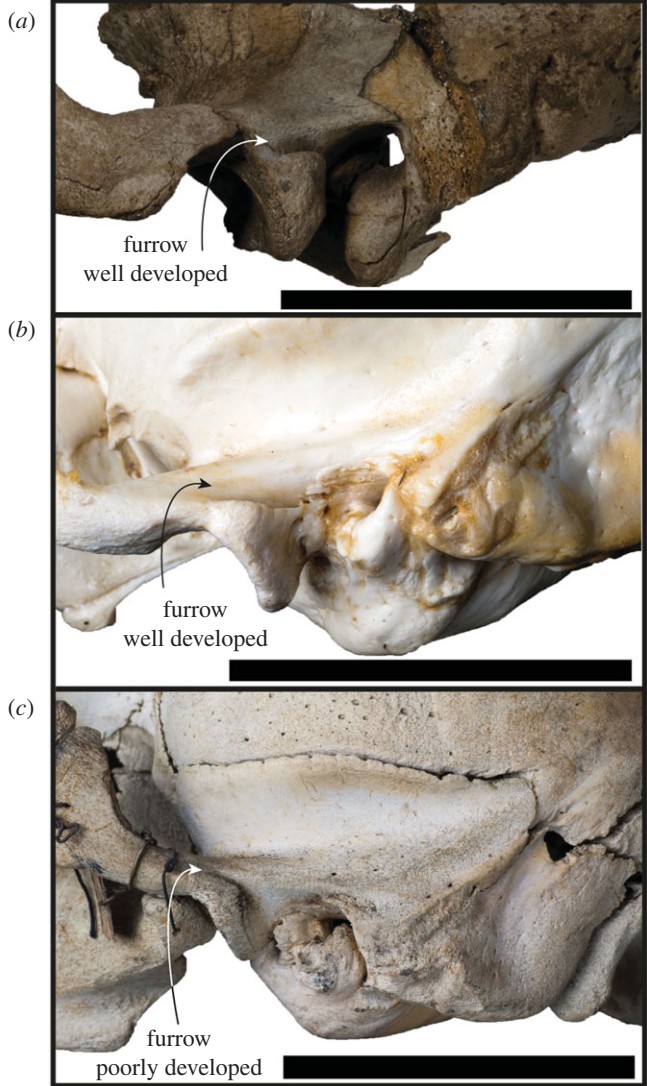

**Figure 13.** Ontogenetic variation of the furrow on the postglenoid. Left lateral views of the braincase of S. magnus USNM PAL 475486 (a), and a skeletally mature adult (NMV C24949, b) and juvenile (NMV C7419, c) of E. barbatus. Scale bars, 5 cm.

cusps of the postcanines of *S. magnus* differ markedly to the carinae present in *H. leptonyx* and the Monachini, but is similar to carinae present on the postcanines of *No. gaudini*. The main and accessory cusps on *H. leptonyx* are tall and cone-like, not low and part of a cutting surface as in *S. magnus* and *No. gaudini* (figures 2*d* and 3*a,b*). The orientation of the postcanines and their cusps are in line with the axis of the tooth row in *S. magnus*, creating a mostly continuous cutting edge. This contrasts with the condition found in other monachines as well as in *No. gaudini*, where the postcanines in adult specimens are oriented at oblique angles to the functional axis of the tooth row. The notched condition between postcanine cusps is noted in some phocids, most notably in *Ac. longirostris* and *No. gaudini.* This intercusp notch condition is very distinct in *Ac. longirostris* compared with *S. magnus* and *No. gaudini*. However, the teeth of *Ac. longirostris* are much more labial-lingually flattened, with the crown elongated along the tooth row with wide diastemata spacing the postcanines.

The upper postcanines are mostly similar to those referred to *Ho. capensis,* with the mesial intercusp notch (between the mesial accessory and main cusps) being reminiscent of the condition of *Homiphoca* lower postcanines. Relative to the dentition of *Homiphoca*, the upper postcanines have a lower main cusp height and greater mesial–distal length. The postcanine crowns are aligned with each other's long axis along the tooth row, in contrast with the en echelon formation of the postcanine tooth rows in monachins, *No. gaudini* and *Ha. martini*. The diastema between the PC4 and PC5 is much larger in *S. magnus* than it is in other fossil monachines, with the exception of *Ac. longirostris*.

The enlarged masseter muscle origin on the jugal of USNM PAL 475486 (figure 2*d*) is not present on any other monachines; consequently, the ventral border of the zygomatic arch does not extend ventral to the

tooth row as it does in crown-lobodontins. The curvature of the ventral edge of the jugal is only seen in *M. monachus*, and is relatively straight in other monachines. Like other Monachinae (and by extension, Phocinae), the orbits of *S. magnus* are relatively large (as a percentage of skull width). When reconstructed, the glenoid fossae are parasagittal in orientation (figure 2), as opposed to the converging state displayed in some monachines (*Mirounga*, *L. weddellii*, *O. rossii* and *Neomonachus*). The squamosal directly dorsal to the glenoid fossa (but anterior to the braincase) is enlarged (but not as enlarged as the condition in *O. rossii*), and is similar to the monachins. The post-auricular shelf is completely absent, a state only shared with *O. rossii* and *Mirounga* (although a reduced state is present in *Ho. capensis*).

The anteromedial apex of the entotympanic (housing the auditory tube) is anteromedially elongated, as it is in *M. monachus* and *Ha. martini* (but not as elongated as it is in *H. leptonyx*). The ectotympanic tubercle is laterally elongated (figures 3*f* and 7), a condition that varies in Monachinae, but is not as exaggerated as it is in *S. magnus* (the exception being *Mirounga* and *O. rossii*, which have unusually elongated ectotympanic tubercles). The tympanohyal fossa is distinct on the posterolateral edge of the bulla, and is not preceded by a groove as it is in other monachines. There is a posterior extension of the tympanic bulla, lateral to the carotid canal but medial to the mastoid. This condition is also present in stem- and crown-lobodontins, as well as miroungins. However, unlike the lobodontins, it is not fully enveloped by a mastoid lip. *Sarcodectes magnus* differs from miroungins in this region as the external cochlear foramen is mostly blocked posteriorly by the contact of the mastoid and the tympanic. The digastric groove is present on the mastoid (figure 7*b*), a structure that leads to the external cochlear foramen in some phocines (ending in the digastric pit in *Pa. groenlandicus* and *Histriophoca fasciata*, see NHMUK 1938.12.10.1 and NHMUK 1965.7.19.8, respectively), and is present in the Monachini and the group of taxa traditionally defined as 'stem-Lobodontini' (*Ac. longirostris*, *Pi. pacifica*, *Ha. martini* and *Ho. capensis*). The mastoid of *S. magnus* contains an enlarged sternomastoid origin (figure 7), similar to monachins and the 'stem-lobodontins'. But it differs from these and other monachines in the morphology of the origin of the digastricus muscle, which is less concave medially, and laterally includes an enlarged mastoid protuberance not seen in other Monachinae (figure 7*b*).

In dorsal view, the medial and anterior petrosal surface is expanded, similar to most Monachinae with the exception of *Neo. schauinslandi* and *Ho. capensis*. However, unlike *Mirounga* and the crown-lobodontins, the petrosal apex is not swollen (similar to *Ho. capensis* and Monachini). The roof of the internal auditory meatus is completely absent (figure 8), with *S. magnus* lacking a remnant of the roof as is present in Monachini. The floccular fossa is open and ovoid in shape (figures 5*c* and 8*b*), similar to all monachines (and phocines), except *Ac. longirostris*, *Neo. schauinslandi* and crown-Lobodontini.

### 3.2.3. Postcranial skeleton

Based on what is preserved, the humerus of *S. magnus* has a deltopectoral crest that gradually attenuates distally (figure 6), a feature shared with all of Monachinae (and the phocine *Kawas benegasorum*). The lateral epicondylar crest is not prominent and does not flare laterally, similar to *Neomonachus*, *O. rossii*, *Mirounga*, *Pi. pacifica*, *Pl. etrusca* and *Au. changorum*. A laterally flaring epicondylar crest is present in crown-lobodontins (except *O. rossii*), *M. monachus*, *Ac. longirostris*, *Ho. capensis*, 'M. ambigua', 'A. atlantica' and 'V. magurai'. The lateral epicondyle of the humerus is not prominent, similar to all Monachinae except *Neo. schauinslandi* and 'A. Atlantica' (figure 14). The medially protruding and rounded medial epicondyle of *S. magnus* is shared with 'M. ambigua' (figure 14), *H. leptonyx, L. weddellii* and an extinct monachine from Beaumaris, Australia [49]. Additionally, the pronator teres origin on *S. magnus* is more proximally located on the medial epicondyle when contrasted to all other Monachinae except *Pi. pacifica* and *L. weddellii* (figures 6 and 15). An antero-posteriorly thickened medial epicondyle is shared with all other monachines except 'M. ambigua' (figure 15) and the Beaumaris humerus [49]. The medial epicondyle of *S. magnus* also lacks the epicondylar foramen that is present in *Ho. capensis* and 'V. magurai'. The virtual lack of an olecranon fossa in *S. magnus* is a feature shared with all Monachinae except 'V. magurai', *Pi. pacifica* and 'M. ambigua'. The medial section of the coronoid fossa being smaller than the lateral section in *S. magnus* is a feature shared with all Monachinae except crown-monachins, crown-lobodontins and *Ac. longirostris*. The extremely shallow radial fossa is shared with all monachines except for *Pi. pacifica*, 'A. atlantica' and the Beaumaris humerus [49].

## 3.3. Body size estimation

Each method of estimation of total body length for *S. magnus* produced varying results, but only the equation for length of auditory bulla (LB) produced an estimate that was below 2.50 m (1.56 m;

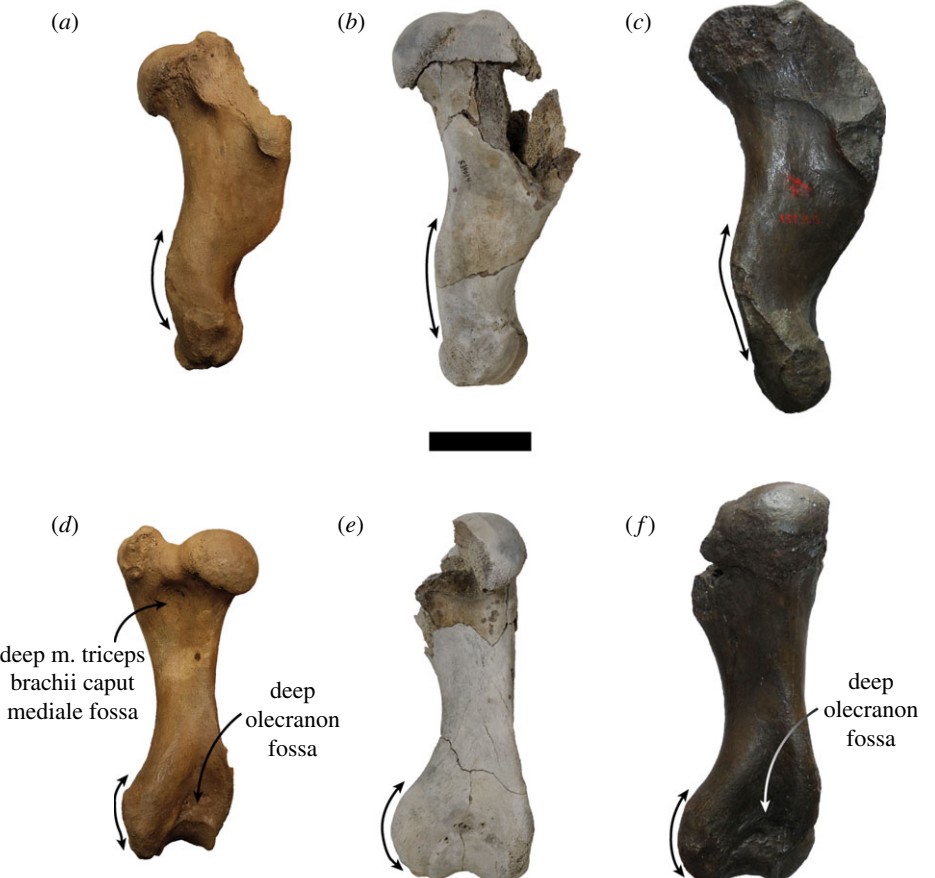

**Figure 14.** Humeri of large extinct Monachinae. The humeri of the type specimens of '*A. atlantica*' USNM PAL 181419 (*a*,*d*), *S. magnus* USNM PAL 534034 and '*M. ambigua*' IRSNB 1156-M177 in lateral (*a*–*c*) and posterior (*d*–*f*) views. (*a*,*c*,*d*,*f*) horizontally flipped. Double-sided arrows indicate bone profile. Scale bar, 5 cm. Photo of IRSNB 1156-M177 taken by Sébastien Beaudart.

table 4). The mean of the estimates for *S. magnus* was 2.83 m. The phocid faunas of both the Yorktown Formation (1.60–2.83 m, 4.9–3.9 Ma) and the Pisco Formation (0.68–2.70 m, 7.6–5.9 Ma) have a larger estimated range (min–max) of total body lengths than the range currently present in the western North Atlantic (1.50–2.70 m, figure 16), indicating there was a greater diversity of body sizes in the past. However, there was substantial overlap in these ranges. This contrasts with the range of total body lengths represented by the monachines of the recent Southern Ocean (2.60–5.00 m), which has minimal overlap with the other three assemblages. The monachines of the recent Southern Ocean are almost all larger than the other three faunas, due to there being no small-sized phocids present (figure 16).

## 3.4. Phylogenetic analysis

The holotype and paratypes of *S. magnus* were coded as a single taxonomic unit. Character codings are listed in electronic supplementary material. The parsimony analysis with equal weights produced four most parsimonious trees, with a tree length of 566 after 22 867 190 rearrangements. A consensus tree (figure 17, consistency index (CI) = 0.355, retention index (RI) = 0.660) resolves a monophyletic Phocinae and crown-Monachinae, with *D. claytoni* being sister to crown Phocidae. '*Leptophoca proxima*' is sister to all other phocines. Phocinae comprises a polytomy at the stem of the crown group between *Erignathus barbatus*, *C. cristata*, *N. vitulinoides* and tribe Phocini. The relationships within Phocini are fully resolved (do not form a polytomy). Monachinae comprises solely a crown group, with two major clades (figure 17). The first clade comprises the Monachini, with *S. magnus* and *Monotherium? wymani* forming a sister clade. *Piscophoca pacifica*, *Ha. martini*, *Ho. capensis* and *Ac. longirostris* (often interpreted as fossil lobodontins; [3]) form a clade sister to the Monachini + *Sarcodectes–Monotherium?* clade. The other clade comprises the Miroungini and Lobodontini (with *O. rossii* and *Lo. carcinophaga*

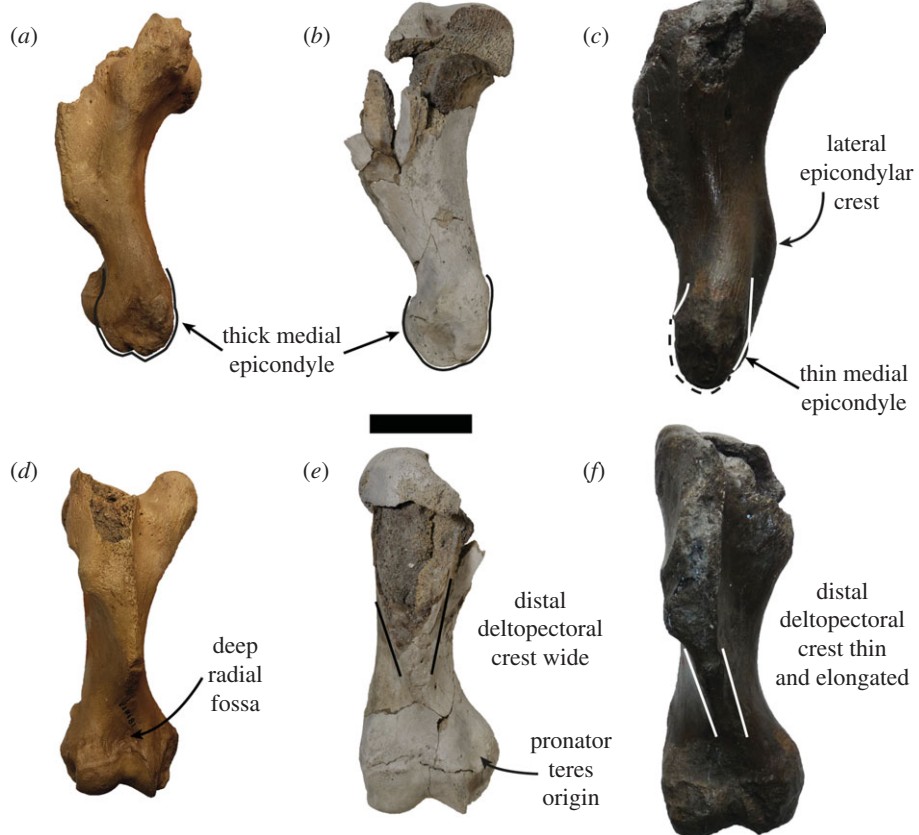

**Figure 15.** Humeri of large extinct Monachinae. The humeri of the type specimens of 'A. atlantica' USNM PAL 181419 (a,d), S. magnus USNM PAL 534034 and 'M. ambigua' IRSNB 1156-M177 in medial (a–c) and anterior (d–f) views. (a,c,d,f) horizontally flipped. Scale bar, 5 cm. Photo of IRSNB 1156-M177 taken by Sébastien Beaudart.

**Table 4.** Body length estimates of holotype and paratype specimens using multiple methods.

| specimen number | estimate method | body length estimate (m) |
|---|---|---|
| USNM PAL 475486 | Churchill *et al.* [25] (LUTR) | 3.47 |
| | Churchill *et al.* [25] (LUPC) | 3.33 |
| USNM PAL 181601 | Churchill *et al.* [25] (WB) | 2.82 |
| | Churchill *et al.* [25] (LB) | 1.56 |
| USNM PAL 534034 | Piérard [26] (*L. weddelli* ratio) | 2.78 |
| | Piérard & Bisaillon [27] (*O. rossi* ratio) | 3.03 |
| mean total length estimate | | 2.83 |

forming a polytomy with *Hydrurga–Leptonychotes*), with *Au. changorum* sister to the Miroungini + Lobodontini clade (figure 17).

Sarcodectes magnus was diagnosed by three apomorphies: an anterior extension of the external acoustic meatus as a deep groove on the postglenoid process (character 64); a tympanohyal fossa present on the tympanic bulla as a shallow pit (character 75); and an enlarged digastric protuberance on the mastoid (character 95).

Clade-specific synapomorphies from the analysis are listed in electronic supplementary material. There were few unequivocal synapomorphies for the major clades within Monachinae, including the novel least inclusive major monachine clade (Clade 1: *Australophoca*, Miroungini, Lobodontini) or the novel most inclusive major monachine clade (Clade 2: Monachini, *Sarcodectes*, *Monotherium?*, *Piscophoca*, *Acrophoca*, *Hadrokirus*, *Homiphoca*). The South American + South African clade (a polytomy

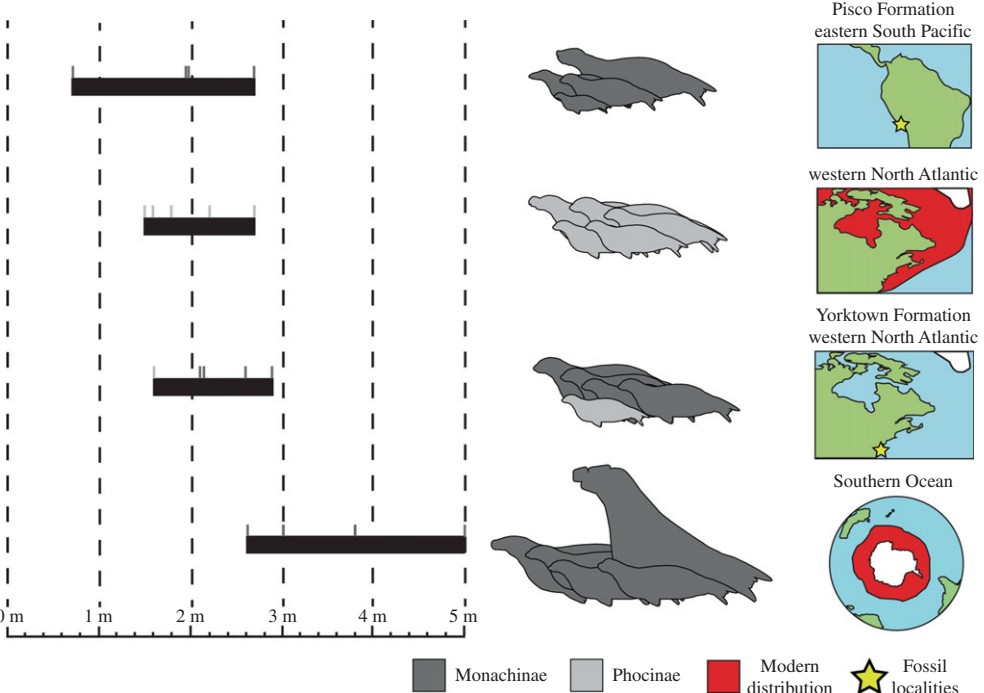

**Figure 16.** Comparison of the maximum total body length among phocid populations. Range of maximum total body length for phocid assemblages around the world. Total body length values and estimations in electronic supplementary material, and indicated by small grey lines above body length ranges. Pisco Formation: *Au. changorum*, *Ac. longirostris*, *Pi. pacifica* and *Ha. martini*. Western North Atlantic: *C. cristata*, *Ha. grypus*, *Pa. groenlandicus*, *Ph. vitulina* and *Pu. hispida*. Yorktown Formation: *S. magnus*, *Homiphoca* sp., '*A. atlantica*', '*V. magurai*' and '*Gryphoca similis*'. Southern Ocean: *M. leonina*, *Lo. carcinophaga*, *O. rossii*, *H. leptonyx* and *L. weddellii*.

of *Piscophoca, Hadrokirus* and *Homiphoca + Acrophoca*) was diagnosed by two unequivocal synapomorphies: parasagittal medial margins of the tympanic bullae (character 67); and a slender, elliptical head of the malleus (character 91).

The maximum-likelihood ancestral state estimation of total body length for Monachinae (figure 18) estimated the node of Monachinae to be 1.97 m (CI: ±0.90). The node of the Miroungini–Lobodontini–*Australophoca* clade (Clade 1) was estimated as 1.89 m (CI: ±1.02; figure 18). The miroungin–lobodontin clade in contrast has a node estimated as 3.34 m (CI: ±0.81). The more inclusive monachine clade (Clade 2) was estimated to be 1.98 m (CI: ±0.82). In most clades, there appeared to be a trend towards larger total body length from the estimated ancestral state, with the only exception being the branch leading to *Au. changorum* where there was a trend towards a smaller total body length (figure 18).

# 4. Discussion

## 4.1. Implications for monachine taxonomy

One of the biggest barriers to progress in phocid taxonomy is sorting through the names assigned to chimeric assemblages of unassociated and fragmentary phocid fossils. A large number of phocid fossils from the Antwerp Basin in Belgium were described early on by Van Beneden [7,50]. These fossils were assigned to names arbitrarily, with no association with each other and no designation of type specimens; this was the same treatment Van Beneden applied to balaenopterid whales [51–55]. For most of the twentieth century, this issue went unaddressed.

Work on the taxonomy of some of these chimeric taxa picked up in the early twenty-first century by Koretsky and co-workers [8,16]. However, minimal investigation into the validity of these taxa has been undertaken; instead, several specimens from the Yorktown Formation in the USA were referred to Van Beneden's chimeric species *C. obscura* from Antwerp [8]. This included the referral of another of Van Beneden's large Antwerp phocids (*M. ambigua*) as the male morph of *C. obscura* [8]. These fossils are almost all isolated, and referred to species without morphological overlap with the type

**Figure 17.** Equal weights consensus parsimony tree. Phylogenetic analysis performed in TNT with equal weights (consensus of four most parsimonious trees). Bootstrap values that exceeded 50% are shown at relevant nodes. Dagger indicates extinct taxa. PLIO, Pliocene; PL, Pleistocene. Supporting information found in electronic supplementary material.

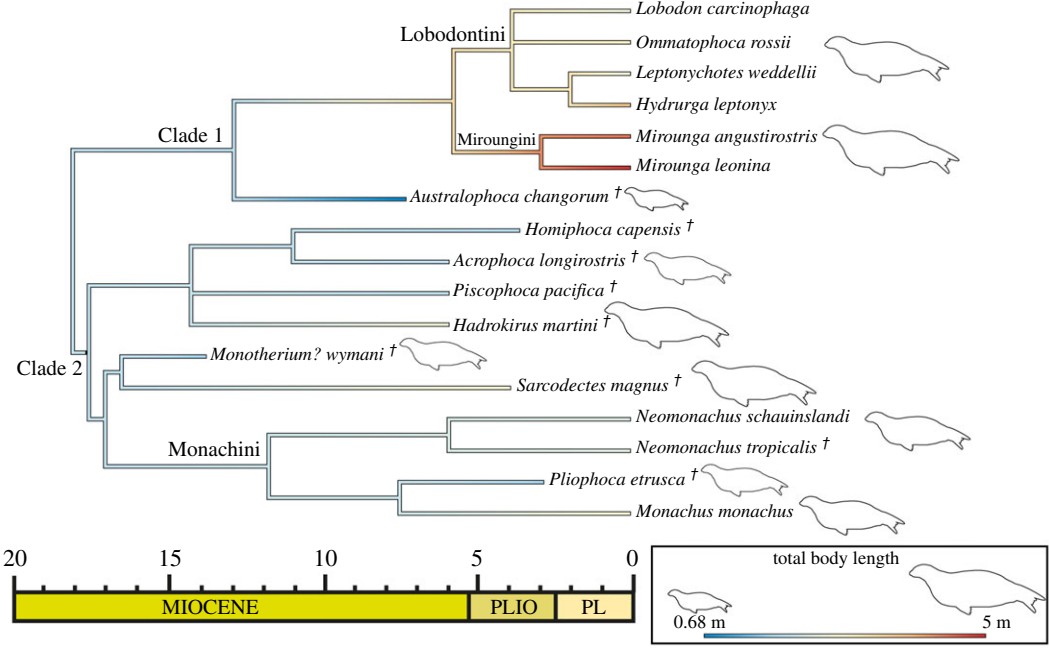

**Figure 18.** Ancestral state estimation of total body length. Maximum-likelihood ancestral state estimation of total body length for the consensus parsimony tree for Monachinae.

**Table 5.** Taxonomic referral of specimens previously referred to *Callophoca obscura*. Anatomical abbreviations: DPC, deltopectoral crest; DT, deltoid tuberosity; LEC, lateral epicondylar crest; ME, medial epicondyle; OF, olecranon fossa; pos., posterior; prox., proximal; PT, pronator teres, RF, radial fossa.

| specimen number | locality | revised referral | anatomical differences |
| --- | --- | --- | --- |
| IRSNB 1156-M177 | Antwerp Basin | *Mesotaria ambigua* | N/A, lectotype |
| IRSNB 1116-M188 | Antwerp Basin | Monachinae indet. | missing: prox. portion and DPC. Thick ME; extreme pos. elongation of LEC; RF and OF absent |
| IRSNB 301 | Antwerp Basin | Monachinae indet. | thin ME; elongated and thin DPC; elongated pos. projecting LEC; RF and OF absent |
| USNM PAL 186944 | Lee Creek Mine | Monachinae indet. | thick ME with angular profile; small LEC; enlarged DT; deep RF; deep OF |
| USNM PAL 263656 | Lee Creek Mine | Monachinae indet. | thick ME with rounded profile; proximal PT origin; LEC not pos. projected; wide DPC; deep RF; deep OF |

specimen. The referral of these unassociated fossils to each other by Koretsky and co-workers is done under the basis of an 'ecomorphotype hypothesis' [16,56]. The 'ecomorphotype hypothesis' states that various bone morphologies belong to 'ecotypes', allowing referral of otherwise unassociated fossils to a type specimen. However, a recent quantitative analysis found no support for this hypothesis [11], and several authors have placed doubts on its validity [9,15,57]. Apart from a loose association of ecology and morphology, the 'ecomorphotype hypothesis' is in essence indistinguishable from the early work of Van Beneden [7,50]. Due to this, it is recommended that the 'ecomorphotype hypothesis' be avoided as a methodology until such a time as it is supported by a rigorous quantitative analysis. Instead, revision of the various phocid species complexes should be done with a more systematic approach [9,15,23], with a focus on the validity of these older type specimens.

As previously highlighted, the incompleteness of the lectotype humerus (figure 1, [9]) means that necessary anatomical features that are critical for comparison and diagnosis of other fossil humeri to *C. obscura* are missing. This, in conjunction with the morphometric study by Churchill & Uhen [11] that doubts the adequacy of isolated humeri as type specimens, justifies scepticism of the validity of *C. obscura*. As noted previously in the comments for the systematic palaeontology, we conclude that *C. obscura* is a nomen dubium, and restrict application of the name to the uninformative and fragmentary lectotype humerus. The benefit of this is the discontinued treatment of *C. obscura* as a so-called wastebasket taxon for large fossil monachines, where large fossil monachine humeri (and other specimens) are referred to this name with minimal basis.

This restriction of *C. obscura* to the lectotype means that the large fossil monachine material from the Antwerp basin in Belgium (possibly Messinian in age, [9]) can no longer be reliably compared with material from the Lee Creek Mine of the USA. Consequently, '*M. ambigua*' is the only large monachine lectotype from the Antwerp Basin. The lectotype specimen of '*M. ambigua*' (IRSNB 1156-M177) has several discernible differences from both the holotypes of '*A. atlantica*' and *S. magnus* (as outlined in the comparison section; figures 14 and 15). As a result, none of the type specimens of these taxa can be reliably referred or synonymized with '*M. ambigua*'. A review of humeri previously referred to *Callophoca* all demonstrate differences from the type specimens of these other large taxa (table 5). As a result, *Sarcodectes* and '*Auroraphoca*' are only known from the Lee Creek Mine, and '*Mesotaria*' is only known from the Antwerp Basin (table 5). Therefore, *S. magnus* is only currently known from the early Pliocene of the western North Atlantic. This further reduces the cosmopolitan nature of fossil seal taxa previously reported from the North Atlantic [8,9].

In the most recent reassessment of Monachinae in the Atlantic, Dewaele *et al.* [9] concluded that only *C. obscura* and *Homiphoca* sp. are known from the eastern and western sides of the Atlantic. As of this study, this is reduced to only *Homiphoca*. This genus is currently only known from Langebaanweg [17,18] and Hondeklip Bay [58] of South Africa, and the Lee Creek Mine of the USA [8,9]. However, all referred specimens from the Lee Creek Mine are humeri, and as such should be considered only tentatively referred to the genus *Homiphoca*. More recently, a fossil innominate bone has been referred to *Homiphoca* from the Zanclean of southwest Spain [59], which would extend the geographical range

of the genus even further. However, this specimen is isolated and fragmentary, and innominate bones have never been considered to be diagnostic to the generic level in prior studies. As such, we recommend this particular specimen be treated only tentatively as *Homiphoca*, until more complete specimens are recovered from the region. Moving forward, we recommend that cranial remains that can be reliably referred to *Homiphoca* should be identified from the USA and Spain, in order to confirm the presence of the genus in those regions.

The description of diagnostic crania and postcrania (especially the humerus) as *S. magnus* enables differentiation of this species from the other local and contemporaneous monachine taxa. Thus, only three valid monachine taxa are known from the Yorktown Formation of the Lee Creek Mine (*Homiphoca* sp., '*A. atlantica*' and *S. magnus*), and five pre-Pleistocene monachine taxa from the western North Atlantic (latter three plus '*V. magurai*' and *Monotherium? wymani*). Two unnamed phocid femora from Charleston, USA, have been suggested to be Chattian (late Oligocene) in age, and hence the oldest record of phocids from the western North Atlantic [60]. Recently, Dewaele *et al.* [15] highlighted that the stratigraphic provenance of these specimens is unclear, and it is likely that they are reworked from overlying Pleistocene beds. The only other locally comparable extinct taxa are the two species of the chimeric genus *Terranectes* [61], which were recently declared nomen dubia [9]. Hence, there are only five confirmed pre-Pleistocene monachines from the western North Atlantic. This revises the diversity of Monachinae in not only the western North Atlantic, but also the Atlantic as a whole.

## 4.2. Functional implications for feeding behaviour

The relationship between morphology and feeding behavior in marine mammals has been the topic of much recent research [1,62–64]. It is, therefore, possible to generate inferences into the possible prey processing behaviour of *S. magnus*, and to eliminate some specific behaviours for which morphology is lacking. Raptorial feeding ([1,65]; sometimes referred to as pierce feeding) has been proposed to have resulted in the simplification of teeth in pinnipeds as a whole [66]. While the majority of phocids are categorized as pierce feeders [67], several phocids display tooth morphology permitting varied feeding behaviors such as filter feeding (*Lobodon* and *Hydrurga*; [2,67]) and the ability to shake or tear flesh from prey (*Hydrurga*; [67,68]). An extreme example is the simplified conical teeth of *Ha. grypus*, which can be used alongside their robust forelimb claws for puncturing and tearing the flesh of large marine mammals [69,70].

Morphological specialization in the postcanines of *S. magnus* suggests some form of enhanced function relative to other phocids. The postcanine tooth row of *S. magnus* preserves in-line carinae with well-developed intercusp notches. Intercusp notches found in terrestrial taxa range from non-existent (e.g. no sharply angled confluence of dental cusps) through to carnassiform, where two carinae closely abut and form a keyhole-like morphology; this morphology is presumed [47] to function as a stress-dissipation structure (see figure 10*a* for an example of a variant of the carnassiform notch in *Acinonyx jubatus*). Increasingly carnassiform intercusp notches are a reliable indicator of lineages trending towards hypercarnivory (see discussion in [47], a diet composed of over 70% vertebrate flesh; [71]).

Morphological specialization of the dentition can facilitate functioning that enhances the ability to process prey items into swallowable pieces. This specialization could evolve secondarily, although it could also represent retention of morphology present in archaic seals. As such, sharp dental morphology can indicate the effectiveness of postcanines in cutting and chewing flesh, as opposed to swallowing smaller prey items whole or tearing/dismembering larger items with the anterior dentition [1,67,68,72,73].

The following traits and detailed morphologies combine to suggest that *S. magnus* may have been able to process vertebrate flesh more effectively than other phocids: the presence of a relatively enlarged zygomatic origin for the masseter muscle (figure 12); the presence of distinct intercusp notches (figure 11, [47,74]); the presence of carinae [73] with postcanines orientated to form a continuous cutting edge (figure 12, [72,74–76]). On the basis of these multiple lines of evidence, we propose that *S. magnus* had an enhanced ability to process the soft tissue of other vertebrates.

There are several morphological indicators that suggest that *S. magnus* would have had a limited ability to shake and tear prey items. Due to the absence of an entepicondylar foramen on the humerus, which is a morphological structure that enables the abduction movement of the forelimb [77], *S. magnus* probably had limited ability to abduct its forelimbs. This probably constrained its ability to use its forelimbs for grasping prey during processing, a behaviour seen in some modern phocine seals when processing large prey [62]. Shaking and tearing behaviour can also be used in conjunction with large conical dentition to process larger prey [68,70]. However, it must be noted that 'larger prey' is relative; while *H. leptonyx* feeds on penguins using a shake and tear method [68], these

are smaller than the size of prey tackled by *Ha. grypus* [62,70]. *Halichoerus grypus* possesses larger, more robust conical dentition in its postcanine tooth row in addition to its canines and incisors, which may assist with 'grasping' pieces of flesh from a larger animal such as a whale [70] or seal [69]. On the other hand, *H. leptonyx* has more robust anterior dentition to assist with shaking and tearing of prey items [68]. The canine of *S. magnus* is relatively short, indicating that shaking and tearing of larger prey using its anterior dentition was possibly limited.

What, then, is the function of these unusual, sharp postcanines? The cutting blades and pronounced intercusp notches present on the postcanine dentition of *S. magnus,* as well as the enlarged origin of the masseter muscle, may have enabled this species to effectively process prey during biting and perhaps repeated bites in chewing, as has been observed in Australian sea lions (*Neophoca cinerea*) [78]. Australian sea lions do not have deep intercusp notches, but do bite and chew in conjunction with shaking. The chewing of prey occurs between bouts of shaking, enabling these sea lions to reduce prey into swallowable pieces. While there is little support for shaking and tearing in *S. magnus*, its various morphological adaptations suggest that there was an enhanced ability for it to bite and chew larger prey to reduce it to a swallowable size. It is difficult to infer what prey *S. magnus* may have been consuming, but a variety of fish [79], turtles [80], birds such as puffins [81,82], dolphins [83,84] and other seals [8,9] are present from the Pliocene of the Lee Creek Mine. Other invertebrate prey, such as squid and octopus, are also likely, based on the diets of modern pinnipeds [85]. The possible ability of *S. magnus* to bite and chew prey using sharp teeth may be similar to what has been hypothesized for the early pinniped *Enaliarctos* [48,62,86,87]. However, due to the limited ability to abduct the forelimb, and the flipper-like morphology of monachines in general [68], *S. magnus* may not have been able to hold it prey using its forelimbs like *Enaliarctos* (and modern phocine seals) during feeding on large prey [62]. As the entepicondylar foramen is present in the fossil monachines *Ho. capensis* and '*V. magurai*', it may be that the ability to abduct the forelimb was retained in archaic monachines (as it was in phocines; [62]), and then subsequently lost. Hence, the potential ability of *S. magnus* to chew and bite prey may reflect a variation of this functional shift in the feeding behavior in extinct Monachinae.

## 4.3. Phylogenetic relationships

The equal weights consensus tree found *D. claytoni* outside crown Phocidae, which was composed of monophyletic Phocinae and Monachinae (figure 17). This supports the traditional division of Phocidae into two subfamilies [3,21,22,29,88–90]. The phylogeny failed to resolve *D. claytoni* as a crown-phocine [23], or as a stem-phocine [12]. The stem position of *D. claytoni* supports the distinction of subfamily Devinophocinae by Koretsky & Holec [91] and Koretsky & Rahmat [92].

The only recent analysis to include the referred skull [16] of *Le. proxima* is Berta *et al.* [10]. While Berta *et al.* interpreted it as a possible stem-monachine, this may be due to the fact that phocines were used as an outgroup. As the skull (CMM-V-2021) has been referred to *Le. proxima* without a way to associate it with the holotype, we treat the skull the same as Dewaele *et al.* [15], and therefore cannot compare our results with the more recent phylogenies in Dewaele *et al.* [15,23]. Until a proper association between the skull (CMM-V-2021) and the holotype can be made, we assert that the skull be treated as its own separate taxonomic unit. The character-rich skull was found as a stem-phocine (figure 17).

The phylogenetic analysis also found *Pl. etrusca* to be sister to *M. monachus* (figure 17), the same relationship estimated by previous studies [3,10,15,20], but contrasting to the sister relationship with a *Monachus* and *Mirounga* clade found by Dewaele *et al.* [23]. *Mirounga* was found to be sister to the lobodontins. Interestingly, *Au. changorum* was found to be sister to the miroungin–lobodontin clade. While *Australophoca* is substantially smaller than the miroungins and lobodontins [6], this relationship is arguably consistent with its geographical position in the South Pacific, as this is close to where the lobodontins (and *M. leonina*) are found. As this is the first phylogenetic analysis to include *Au. changorum*, follow-up studies should seek to test this hypothesized relationship through character analysis.

The analysis did not support previous findings of *Homiphoca*, *Piscophoca*, *Hadrokirus* and *Acrophoca* as stem-lobodontins [3,10,15,20], instead clustering them in an entirely different clade (Clade 2, figure 18). This also contrasts with the most recent analysis that found them as stem-monachines [23]. This quartet of genera has long been implicated in the origins of crown-Lobodontini, with a dispersal south from the eastern South Pacific into the Southern Ocean [28,29,89]. This is mostly based on their auditory region, with a posterior extension of the tympanic and an unexposed external cochlear foramen, uniting these taxa with Lobodontini [3,90]. Several of these characters were redefined in this study; an additional state was added to each character. The posterior extension of the tympanic character of Amson and de Muizon [3] was modified to account for differences in its profile between modern phocines and

monachines (character 82; electronic supplementary material). An additional state to the mastoid lip character of Amson & de Muizon [3] was included to reflect a condition where the mastoid covers the tympanic extension dorsally, but a lip does not abut the bulla (character 83; electronic supplementary material) which is present in the extant *O. rossii*, and the extinct *Monotherium? wymani* and *S. magnus*. The external cochlear foramen character of Berta *et al.* [10] was completely redefined (character 84; electronic supplementary material) to reflect the influence of the mastoid–tympanic contact (or lack thereof). The reassessment of these tympano-mastoid characters may indicate that the mastoid lip ear morphology is more prevalent in extinct Monachinae than previously thought.

*Sarcodectes magnus* was found to form a clade with *Monotherium? wymani*, which was sister to the Monachini (figure 17). An early analysis by de Muizon [89], as well as some exploratory analyses by Dewaele *et al.* [23], indicated that *Monotherium? wymani* was a lobodontin. However, as outlined above, the specific ear morphologies previously assumed to be exclusive to Lobodontini may be more widespread in Monachinae than previously thought. The only studies to include the holotype of *S. magnus* are Amson & de Muizon [3] and Berta *et al.* [10], which coded the specimen in the 'C. obscura' complex (it is uncertain, but assumed, that this specimen was also included in the analyses of Koretsky & Holec [91] and Koretsky & Rahmat [92]). However, as already highlighted here, as well as in Berta *et al.* [10] and Dewaele *et al.* [9], this holotype material of *S. magnus* had a dubious referral to 'C. obscura'. While the results of these studies are non-comparable with the results here, it should be noted that Amson and de Muizon [3] found 'C. obscura' to be sister to *Mirounga*, and Berta *et al.* [10] found 'C. obscura' to be a stem-monachine. Future phylogenetic analyses, with a continued conservative approach to operational taxonomic units, will allow better exploration of the relationships of *S. magnus* beyond this study.

The phylogenetic hypothesis presented here has implications for what we know regarding monachine palaeobiogeography. The stem relationship of *Sarcodectes* and *Monotherium?* to the Monachini suggest a North Atlantic origin for this clade, which supports what has recently been proposed [10,29,93]. However, the stem taxa of both clades 1 and 2 (figures 17 and 18) all come from the eastern South Pacific (with the exception of *Ho. capensis*). This would instead suggest an origin, or at least a dispersal, of the *Sarcodectes*–*Monotherium?* + Monachini clade via the Central American Seaway into the Atlantic. In addition, this hints at a Pacific origin for crown-Monachinae. This contrasts greatly with the biogeographic dispersal patterns that have previously been suggested [29,94]. This is especially important considering the phylogenetic and biogeographic analyses of both Flynn *et al.* [94] and Fulton & Strobeck [29] based their calibrations and dispersal routes on the assumed relationship of 'Callophoca' with *Mirounga*. Several southern fossil monachines are known from the Neogene of the Southern Hemisphere [95–97], as well as from the Neogene of the North Pacific [98]. However, these fossils, as well as stem-monachines, are not included in this study, and a rigorous biogeographic analysis would require a phylogeny including such fossils, as well as accurate estimates of divergence times. As such, the description of *S. magnus* calls for a new biogeographic analysis of the Monachinae.

## 4.4. Body size evolution

While various body length estimates were found for *S. magnus*, the mean total length was 2.83 m (table 4), making it the largest known extinct monachine. The description of the Lee Creek material, and its inclusion in a phylogenetic analysis, means that the evolution of body size can be assessed using a phylogenetic framework without chimeric taxa (from sites with multiple phocid species present). The only comparable large-scale study into body size including fossil phocids was done by Churchill *et al.* [5]. Apart from methodological differences, their study included two extinct species based on chimeric composite names (*C. obscura* and *Leptophoca lenis*). In addition, the only fossil taxa in the composite phylogeny used were stem-monachines (differing from our phylogeny with only crown monachines). Nevertheless, the results for Monachinae as a whole are broadly comparable with our analysis.

Churchill *et al.* [5] estimated that the common ancestor of Monachinae was approximately 2 m in length, similar to our estimate of approximately 1.97 m. Overall, they found that monachines appear to increase in size through time (following Cope's rule). This is not strictly supported by our maximum-likelihood ancestral state estimation of total body length, as there is a reversal towards a smaller total body length for *Au. changorum*, an extinct monachine (figure 18). However, our results are similar to Churchill *et al.* [5] in that the modern taxa appear to have increased in total body length from the ancestral estimate for Monachinae (figure 18). The biggest estimated increase in total body length is in the Lobodontini–Miroungini clade. It could be that the changing ocean environment after the onset of the Pleistocene glaciations [99] may have enabled larger body size to be more broadly

attained in Monachinae. This may specifically be linked with changes in primary productivity, as indicated by Churchill *et al.* [5]. The trends presented here should also be noted to be dependent on the topology of the phylogeny used. Further work will help explore the trends in body size evolution in Monachinae.

The largest phocid species from the Yorktown Formation (*S. magnus*) approaches the size of the walrus *Ontocetus emmonsi*, which is also represented in the Yorktown Formation [5,100,101], indicating *Sarcodectes* may not have been the largest pinniped present in the region. When comparing the Yorktown Formation with other phocid assemblages, the western North Atlantic fauna of the early Pliocene was much more diverse in total body length than the exclusively phocine fauna that exists there in more recent times (figure 16). Even more diverse is the slightly older late Miocene fauna of the eastern South Pacific (figure 16). The presence of the small-sized *Au. changorum* [6] demonstrates that there was at least some size diversity in early Monachinae. All of these faunas contrast greatly to the diversity of total body length in the recent Southern Ocean (figure 16). This body length diversity is largely skewed by the large size of male *M. leonina* within the fauna, but still represents a phocid fauna of a size not achieved elsewhere in the past or the present. However, the Southern Ocean fauna has been noted to occupy a similar range of sizes to extinct Neogene pinniped faunas of the eastern North Pacific [6]. As the recent Southern Ocean phocids assemblage is composed of members of the Lobodontini–Miroungini clade, the Southern Ocean may have been important for increases in body size for phocids in the past. However, this pattern may simply reflect the fact that the majority of recent monachine diversity exists in the Southern Ocean.

In regard to sexual dimorphism, fossils previously referred to '*C. obscura*' were postulated to represent sexually dimorphic males and females [8,9]. There are no other known fossil phocids that display sexual dimorphism. Some of the monachine humeri previously referred to '*C. obscura*' from both the Lee Creek Mine of the USA [8] are larger than the paratype humerus of *S. magnus* (USNM PAL 534034) by approximately 4 cm. However, this may not be an indication of dimorphism, as the paratype humerus has unfused proximal epiphysis, which may indicate that the maximum size of that individual had not been obtained. As a result, there are no currently known sexually dimorphic extinct phocids.

# 5. Conclusion

With *C. obscura* being found to be a nomen dubium and restricted to the lectotype, we conclude that the Yorktown Formation material should be described as a new taxon, *S. magnus*. This taxon demonstrates morphological adaptations for processing the flesh of prey that are unique to Phocidae. This taxonomic reassessment enables the reliable inclusion of the large Yorktown Formation monachine into phylogenetic analyses of Phocidae. Our phylogenetic analysis found *S. magnus* to form a clade with *Monotherium? wymani*, which is a sister clade to Monachini. The description of this taxon, and the phylogenetic analysis presented here, allows for the evolution of body size in Monachinae to be put into context. Cope's rule is not supported for Monachinae, due to the size decrease in the evolution of *Australophoca*; large monachines do not appear to have evolved until the diversification of the lobodontins and miroungins. While the range of total body length during the early Pliocene of the eastern North Atlantic was more diverse than today, it did not approach the range present in the recent Southern Ocean. The description of *S. magnus* will allow an improved assessment of the diversity of body size within Monachinae in future studies.

Data accessibility. All data for this paper are either contained within this paper, its electronic supplementary material or have been uploaded to Morphosource. Measurements of specimens, additional figures, body length estimation data, phylogenetic character list, character codings and additional data for the parsimony phylogenetic analysis can be found in the electronic supplementary material. The three-dimensional reconstruction of USNM PAL 475486 can be found at doi:10.17602/M2/M116091. This published paper and the nomenclatural acts within are registered with ZooBank. This publication's LSID is urn:lsid:zoobank.org:pub:97AF4D81-749F-409A-A0AD-7D76BF3B310E.
Authors' contributions. J.P.R. designed the study; collected the data; carried out descriptions, comparisons, measurements, body length estimations, phylogenetic analysis and ancestral state estimation; contributed to the discussion; wrote the paper. J.W.A. carried out descriptions and body length estimations; contributed to the discussion; wrote the paper. D.S.R. carried out descriptions, comparisons, phylogenetic analysis; contributed to the discussion; wrote the paper. D.P.H. contributed to the discussion; wrote the paper. A.R.E. contributed to the discussion; wrote the paper. E.M.G.F. designed the study; carried out the phylogenetic analysis; contributed to the discussion; wrote the paper. All authors read the final version of this manuscript and approved it publication.
Competing interests. We declare we have no competing interests.
Funding. J.P.R. is supported by the Australian Government Research Training Program (RTP) stipend scholarship, and D.S.R. is supported by the Department of Anatomy and Developmental Biology at Monash University. Both J.P.R. and

D.S.R. are supported by a Robert Blackwood Partnership Monash–Museums Victoria scholarship. D.P.H. was supported by Australian Research Council DP18010179. Monash Biomedical Discovery Institute and a Monash University Graduate Research Travel Grant funded travel to visit museum collections for this project.

Acknowledgements. The research undertaken in this paper forms part of the PhD research for J.P.R. at Monash University, with J.W.A., A.R.E. and E.M.G.F. as supervisors. The following museums and staff are thanked for access to collections: Museums Victoria (T. Ziegler, K. Roberts, K. Date, R.-L. Erickson, K. Rowe); the Natural History Museum London (R. Miguez, P. Jenkins); Muséum national d'Histoire naturelle (C. de Muizon, G. Billet); Smithsonian Institution National Museum of Natural History (J. Mead, J. Ososky, M. McGowen, D. Lunde, D. Bohaska, N. Pyenson); Calvert Marine Museum (S. Godfrey); Institut royal des Sciences naturelles de Belgique (A. Folie); and the Natural History Museum of Los Angeles County (J. Velez-Juarbe). Thanks are given to S. Beaudart, who provided photos of IRSNB 1198-M203 and IRSNB 1156-M177. T.I. Pollock is thanked for assistance with the R package 'Geiger'. The following collectors are thanked for the discovery and donation of specimens: C. Swindell (USNM PAL 475486); F.V.H. Grady (USNM PAL 534034) and P.J. Harmatuk (USNM PAL 181601). Thanks to the editors (Denise Greig and Kevin Padian) and the reviewers (Robert Boessenecker and an anonymous reviewer) for comments that improved this manuscript.

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
