## [Reviewer comments · Royal Society Open Science]

Review History

RSOS-200796.R0 (Original submission)

Review form: Reviewer 1 (Robert Boessenecker)

Is the manuscript scientifically sound in its present form?

No

Are the interpretations and conclusions justified by the results?

Yes

Is the language acceptable?

Yes

Do you have any ethical concerns with this paper?

No

Have you any concerns about statistical analyses in this paper?

No

Recommendation?

Major revision is needed (please make suggestions in comments)

Comments to the Author(s)

This manuscript is a competent addition to the literature on fossil seals, and provides a much needed, albeit limited, revision of the large bodied Pliocene seal *Callophoca obscura* from the North Atlantic. *Callophoca* was originally named based off of a fairly eroded but large humerus missing most of the anatomically informative proximal and distal ends, collected from the Pliocene of Belgium. Van Beneden assigned a “type series” of non-associated fossils to *Callophoca*, and a lectotype humerus was designated by Koretsky and Ray (2008). This latter study was a bit of a dog’s breakfast, repeating many of the same mistakes as Van Beneden – assigning nonassociated fossils from the Lee Creek Mine in North Carolina to a morphotype based upon a preconceived notion of what the different parts of the seal “should” look like, versus emphasizing consideration of overlapping parts only. One of the specimens, referred originally on size, is a partial skull, here designated as the holotype of *Sarcodectes magnus*. This new species is competently described with particular attention towards interpreting its feeding ecology and the evolution of body size in monachine seals.

However, there is a serious taxonomic problem buried within this. The lectotype specimen of *Callophoca obscura*, IRSNB 1198, is not informative, and the authors are indeed well-supported in their bid to designate it a nomen dubium. However, the path forward to name a new species or genus is far from clear, and aside from tossing out *Callophoca*, little attention is paid in the manuscript to other Belgian fossil phocids, which are sort of sidestepped. Koretsky and Ray (2008) also designated a lectotype for *Mesotaria ambigua*, IRSNB 1156, which is a nearly complete humerus missing only the tip of the lesser tuberosity. This humerus, figured quite nicely by PJ Van Beneden 1877: plate 9.9-9.11, is nearly identical to the humerus of USNM 534034, referred in this manuscript to *Sarcodectes magnus* and linked to the proposed holotype cranium by a squamosal. The *Mesotaria ambigua* lectotype, unlike that of *Callophoca obscura*, is diagnostic and the same arguments for a nomen dubium cannot be made. Furthermore, Koretsky and Ray (2008) declared *Mesotaria ambigua* a synonym of *Callophoca obscura* and referred the lectotype humerus to *Callophoca*, so there is existing published opinion that the *Mesotaria ambigua* lectotype humerus and the proposed *Sarcodectes magnus* holotype and referred specimens are part of the same hypodigm. A second even better preserved humerus exists, figured by Koretsky and Ray (2008: figure 24a, 25a, 26a, 27a) and referred to *Mesotaria* by Van Beneden 1876 but not figured. In my opinion, caution is warranted and at present I think a case for a new genus/species is not strong, particularly given that no mention of *Mesotaria ambigua* is made in the present manuscript. Owing to the similarity of the *Mesotaria ambigua* lectotype with USNM 534034, this taxon, owing to taxonomic priority, is probably better referred to as *Mesotaria ambigua* or perhaps a new species of *Mesotaria* (*Mesotaria sarcodectes*, perhaps? That way you could keep that killer name). Detailed comparisons have yet to be made and are a starting point for addressing this issue.

Otherwise, I am very impressed with the current manuscript and I commend the level of detail, and have a number of moderate and minor suggestions that could improve the long term utility of the work.

I commend Mr. Rule on his excellent work and have no concerns that he and the rest of the team will be able to take an already great paper and make it into a fantastic one. Kind regards, Robert Boessenecker

Moderate comments

1) As mentioned above, Belgian specimens/taxa aside from *Callophoca obscura* are sort of side-stepped. A further issue is that USNM 534034 is not the only large monachine humerus from the Lee Creek Mine, and there are others figured by Koretsky and Ray (2008). Are these, or are they not, the same taxon? For example: USNM 186944, 263656. Further to the point, the authors claim that *Mesotaria/Sarcodectes* is known only from the Western North Atlantic, but have not bothered making any comparisons with Belgian specimens of “*Callophoca*” referred/figured by Koretsky and Ray that happen to be quite well preserved (e.g. IRSNB 301).

- 2) Please consider including the holotype skull measurements as a table within the main text.
- 3) Why not compare the range of body sizes, at a faunal level, with those from the North Pacific? Also, regarding pinniped assemblages as a whole, it may be worth mentioning that there is a much larger bodied walrus (*Ontocetus emmonsii*) in the North Atlantic – a bit odd to leave out of a faunal discussion. Also, as regards the extant Antarctic assemblage, some explanation/hypotheses for why there are larger body sizes across the board would be welcome – higher primary productivity, colder water temperatures, perhaps? Some combination?
- 4) It would be wise to include a figure of the *Callophoca obscura* lectotype humerus IRSNB 1198, in the interest of transparency.
- 5) In various places ‘large’ and ‘small’ are used in the absence of quantification; please include brief quantifiers in the text (e.g. small, <10% of bizygomatic width).
- 6) Some more synapomorphies should be listed in the main text, as written the inclusion of synapomorphies is a bit lopsided. Better yet, including them on an annotated cladogram would really help – just within the Monachinae, anyway, or maybe even just the Monachini + Mesotaria/Sarcodectes + Peruvian seal clade. At minimum, select synapomorphies for the Mesotaria/Sarcodectes + ?*Monotherium* + Monachini clade and Mesotaria/Sarcodectes + ?*Monotherium* clade should be included in the main text, since these are novel clades (if memory serves).
- 7) It hasn’t really been discussed much in the pinniped literature, but Van Beneden did the same thing with phocids that he did for whales: assembling chimaeric assemblages and not naming lectotypes, causing taxonomic paralysis and/or confusion for balaenopterid whales, cetotheriids, cetotheres sensu lato, and phocids for over a century. A bit of this, using *Herpetocetus scaldiensis*, is laid out by Demere et al. (2005) and El Adli et al. (2012). A brief discussion of this practice – and the apparent and unfortunate continuation of it by Koretsky et al. in the past 20 years – is warranted, particularly now that the Koretsky et al. “analysis” of “ecomorphotypes” has finally been “published” (Koretsky et al. 2020, First description of ecomorphotypes in seal subfamilies, *International Journal of Zoology and Animal Biology*). Nobody has said it yet, but Koretsky’s methods are essentially the same as Van Beneden’s imaginary seal constructs, just with a fancy name attached. This is a suggestion, but such an indictment is far from unwarranted.
- 8) On line 585-585 the authors cast doubt on the referral of an inominine from Spain to *Homiphoca* “given the various apparent morphotypes of *Homiphoca* from Langebaanweg...”, but on line 80-81 the authors state that “an initial study indicated there may be multiple species of seal...follow up work failed to support this...” and “as a result we have coded both crania and postcrania referred to *Homiphoca capensis* from Langebaanweg.” Is this a double standard being applied or a vestige of an earlier draft? If Rahmat et al. were right, and the Spanish specimen is *Homiphoca*, then credit is due.
- 9) Please include humeral features in the diagnosis. Diagnoses can be constructed at a hypodigm framework and should not be limited strictly to the holotype.
- 10) Lines 759-763: There are two issues here needing attendance. 1) Dewaele et al. 2018 reported a second large monachine seal from the Pliocene of Belgium, about the size of “*Callophoca*”. 2) “Whether this represents *Sarcodectes*...or if some of the Lee Creek Mine material is truly synonymous...” Shouldn’t this question be answered in this paper? As written, the authors seemingly leave the door open to *Callophoca obscura* possibly being diagnoseable after all.
- 11) Why do your results differ in some ways from the Churchill et al. results?

12) I'd like to see a couple of additions to the figures – a complete image of the cranium in lateral view, maybe mirroring the right side of the rostrum with the left braincase/zygoma; a line drawing reconstruction in dorsal/latera/ventral would be an additional nice touch. For figure 7, enlarged views of all the teeth could easily be added onto the right side of the figure, blown up to 4x or so the size in the existing figures. As a matter of fact, there aren't many informative photos of the distinctive teeth (that so much of the text is dedicated to describing and interpreting) provided in the manuscript. The Western North Atlantic maps in Figure 13 are so zoomed in and cropped that it's difficult to properly interpret them – and those weird blobs for the fossil distribution – why not just do “x marks the spot”?

Minor comments

81: delete “work” or “study”

105: What exactly are the preferred methods? Which skull measurements do they use? It's a little ambiguous as written.

187: “inserted between the frontal”... frontals are being referred to collectively here (between) and should be plural

193: please quantify what low in height means

197: please quantify what large means

211: please include the collector's name here; ditto on line 217 for the referred specimen.

226: What member of the Yorktown Formation are these from?

228: Since that text is buried in a supplementary info document, you should probably just go ahead and cite the original sources here.

296: delete “is” after relatively

298: please quantify orbit size, perhaps as a % of skull width, and maybe present some comparisons with other phocids

305: please clarify “post auricular surface”. Are you talking about the squamosal fossa? That's generally what I've heard it termed.

357: Could you please quantify the angle that the tooththrows diverge at?

361: carina should be pluralized as carinae.

367: shelf on the cingulum?

370: “separated from PC4.”

373: ‘distinct’ may be overused in this text.

391: “underdeveloped” is certainly overused in this text and is not informative; better to say “fossae are shallow.”

510: underdeveloped again

511: “is shared with Hydrurga”

523: Based on which taxa from the Yorktown Formation?

523: Please clarify 'larger estimated range of total body lengths' – mean or max/min?

537: "the phocins are resolved" How are they resolved? This is pretty vague. They're not the focus of the current work, so you can get away with "resolved similar to analysis X or Y..."

592: As written, it sounds tentative and that you're about to lead into a discussion of comparisons, but that's already taken place; the next sentence goes into the infamous Oligocene specimens from Charleston. I suggest rewriting it to reiterate your earlier conclusions.

596: It would be good to make some reference to "Terranectes", which is certifiably a chimaera, but parts thereof may still represent some kind of a monachine.

624: Enlargement of the posterior oral cavity, and a shorter, wider rostrum/palate are generally associated with suction feeding (e.g. Werth, 2006, J. Mammalogy; Boessenecker et al., 2017: Proc B), so I am confused why this is evidence for raptorial feeding. It's also not really discussed again. Suggest clarifying or deleting.

648: Does Neophoca have sharp carinae? My impression is "not really".

654: There are also small dolphins present, like cf. Pontoporia and Auroracetus.

663: suggest 'extinct' rather than 'past'

670: This is somewhat unfair; Koretsky and Holec named the clade (we didn't, and our 2018 review paper is not an in depth taxonomic/phylogenetic evaluation), and they even performed a fairly bare-bones analysis. Much of Koretsky's work deserves skepticism but credit is needed where it is due, and at least a couple of analyses have confirmed her result. Perhaps frame it like "initial rudimentary phylogenetic hypotheses supported a stem-phocid position for Devinophoca (Koretsky and Holec, 2002), though later recovered as a stem phocine (Dewaele et al. 2017). We're all flattered of course but I think our phylogeny is based off of Fulton and Strobeck (2010), which has its own problems, and ours was a composite phylogeny (like those of Demere et al. 2003).

683: Use something else other than "a lot"

707-708: I think it's usually just cited as Muizon rather than de Muizon in text

713: our (Boessenecker and Churchill's) 2016 paper on the Waipunga seal from NZ also cast doubt on Miroungin affinities of Callophoca aside from "it's a big seal".

718: "recently been put forward" a bit unclear/awkward

721: Typo – Sacroductes

723: "put forward" again

723: what types of analyses? As written it assumes the reader is already familiar with these analyses

744-749: add some comparison measurements here

750: size of the fauna or body sizes within the fauna?

768-769: This is misleading, as the skull of Mesotaria/Sarcodectes/Callophoca has been coded in prior analyses. As written, the reader could mistake this for meaning that phylogenetic analysis of this specimen was attempted here for the first time – which is not true.

Figure 3 caption: 'ear region' could probably be replaced with squamosal.

Figure 13: please list the taxa, either on the image, or in the caption, and maybe provide the individual data points on the graph on the left, above the bars, for clarity.

Review form: Reviewer 2 (Márton Rabi)

Is the manuscript scientifically sound in its present form?

Yes

Are the interpretations and conclusions justified by the results?

Yes

Is the language acceptable?

Yes

Do you have any ethical concerns with this paper?

Yes

Have you any concerns about statistical analyses in this paper?

Yes

Recommendation?

Accept with minor revision (please list in comments)

Comments to the Author(s)

Very clear manuscript. In the abstract, please constrain the timing of body size increase. I suggest including the body-size analysis into the title. It is a more important contribution of the study compared to the report of unusual dentition. Besides, "unusual dentition" alone is not informative. I also suggest changing the title of the section "Functional implications" to something more informative and specific.

Questions I suggest addressing:

1. The authors added new states to several characters, which had an impact on the phylogeny. Have the authors considered whether these multistate characters can be ordered? If the states form a morphocline and ordered that may impact their analysis. What is the reason for unordering all characters? Multistate characters that can be organized to form a morphocline should be ordered otherwise why forming a multistate character in the first place in those cases?
2. To which extent previous alternative phylogenies would change the overall pattern found by the body size analysis of the current study?

Decision letter (RSOS-200796.R0)

Dear Mr Rule:

Manuscript ID RSOS-200796 entitled "A new large-bodied Pliocene seal with unusual cutting teeth" which you submitted to Royal Society Open Science, has been reviewed. The comments from reviewers are included at the bottom of this letter.

In view of the criticisms of the reviewers, the manuscript has been rejected in its current form. However, a new manuscript may be submitted which takes into consideration these comments.

Please note that resubmitting your manuscript does not guarantee eventual acceptance, and that your resubmission will be subject to peer review before a decision is made.

Your resubmitted manuscript should be submitted by 12-Jan-2021. If you are unable to submit by this date please contact the Editorial Office.

on behalf of Dr Denise Greig (Associate Editor) and Kevin Padian (Subject Editor)
openscience@royalsociety.org

Associate Editor Comments to Author (Dr Denise Greig):

Thank you for this interesting submission. Both reviewers were enthusiastic about your manuscript and one of the reviewers provided extensive and useful suggestions for strengthening your text and figures. I look forward to reading a revised submission.

Editor comments:

Thanks for your submission. Given the extensive recommendations it may take more than our standard three weeks to revise carefully, so this partly informs our reject/resub decision. Please contact the office if you have any questions, and best wishes.

Reviewers' Comments to Author:

Reviewer: 1

Comments to the Author(s)

This manuscript is a competent addition to the literature on fossil seals, and provides a much needed, albeit limited, revision of the large bodied Pliocene seal *Callophoca obscura* from the North Atlantic. *Callophoca* was originally named based off of a fairly eroded but large humerus missing most of the anatomically informative proximal and distal ends, collected from the

Pliocene of Belgium. Van Beneden assigned a “type series” of non-associated fossils to *Callophoca*, and a lectotype humerus was designated by Koretsky and Ray (2008). This latter study was a bit of a dog’s breakfast, repeating many of the same mistakes as Van Beneden – assigning nonassociated fossils from the Lee Creek Mine in North Carolina to a morphotype based upon a preconceived notion of what the different parts of the seal “should” look like, versus emphasizing consideration of overlapping parts only. One of the specimens, referred originally on size, is a partial skull, here designated as the holotype of *Sarcodectes magnus*. This new species is competently described with particular attention towards interpreting its feeding ecology and the evolution of body size in monachine seals.

However, there is a serious taxonomic problem buried within this. The lectotype specimen of *Callophoca obscura*, IRSNB 1198, is not informative, and the authors are indeed well-supported in their bid to designate it a nomen dubium. However, the path forward to name a new species or genus is far from clear, and aside from tossing out *Callophoca*, little attention is paid in the manuscript to other Belgian fossil phocids, which are sort of sidestepped. Koretsky and Ray (2008) also designated a lectotype for *Mesotaria ambigua*, IRSNB 1156, which is a nearly complete humerus missing only the tip of the lesser tuberosity. This humerus, figured quite nicely by PJ Van Beneden 1877: plate 9.9-9.11, is nearly identical to the humerus of USNM 534034, referred in this manuscript to *Sarcodectes magnus* and linked to the proposed holotype cranium by a squamosal. The *Mesotaria ambigua* lectotype, unlike that of *Callophoca obscura*, is diagnostic and the same arguments for a nomen dubium cannot be made. Furthermore, Koretsky and Ray (2008) declared *Mesotaria ambigua* a synonym of *Callophoca obscura* and referred the lectotype humerus to *Callophoca*, so there is existing published opinion that the *Mesotaria ambigua* lectotype humerus and the proposed *Sarcodectes magnus* holotype and referred specimens are part of the same hypodigm. A second even better preserved humerus exists, figured by Koretsky and Ray (2008: figure 24a, 25a, 26a, 27a) and referred to *Mesotaria* by Van Beneden 1876 but not figured. In my opinion, caution is warranted and at present I think a case for a new genus/species is not strong, particularly given that no mention of *Mesotaria ambigua* is made in the present manuscript. Owing to the similarity of the *Mesotaria ambigua* lectotype with USNM 534034, this taxon, owing to taxonomic priority, is probably better referred to as *Mesotaria ambigua* or perhaps a new species of *Mesotaria* (*Mesotaria sarcodectes*, perhaps? That way you could keep that killer name). Detailed comparisons have yet to be made and are a starting point for addressing this issue.

Otherwise, I am very impressed with the current manuscript and I commend the level of detail, and have a number of moderate and minor suggestions that could improve the long term utility of the work.

I commend Mr. Rule on his excellent work and have no concerns that he and the rest of the team will be able to take an already great paper and make it into a fantastic one. Kind regards, Robert Boessenecker

Moderate comments

1) As mentioned above, Belgian specimens/taxa aside from *Callophoca obscura* are sort of side-stepped. A further issue is that USNM 534034 is not the only large monachine humerus from the Lee Creek Mine, and there are others figured by Koretsky and Ray (2008). Are these, or are they not, the same taxon? For example: USNM 186944, 263656. Further to the point, the authors claim that *Mesotaria/Sarcodectes* is known only from the Western North Atlantic, but have not bothered making any comparisons with Belgian specimens of “*Callophoca*” referred/figured by Koretsky and Ray that happen to be quite well preserved (e.g. IRSNB 301).

2) Please consider including the holotype skull measurements as a table within the main text.

3) Why not compare the range of body sizes, at a faunal level, with those from the North Pacific? Also, regarding pinniped assemblages as a whole, it may be worth mentioning that there is a

much larger bodied walrus (*Otocetus emmonsii*) in the North Atlantic – a bit odd to leave out of a faunal discussion. Also, as regards the extant Antarctic assemblage, some explanation/hypotheses for why there are larger body sizes across the board would be welcome – higher primary productivity, colder water temperatures, perhaps? Some combination?

4) It would be wise to include a figure of the *Callophoca obscura* lectotype humerus IRSNB 1198, in the interest of transparency.

5) In various places ‘large’ and ‘small’ are used in the absence of quantification; please include brief quantifiers in the text (e.g. small, <10% of bizygomatic width).

6) Some more synapomorphies should be listed in the main text, as written the inclusion of synapomorphies is a bit lopsided. Better yet, including them on an annotated cladogram would really help – just within the Monachinae, anyway, or maybe even just the Monachini + Mesotaria/Sarcodectes + Peruvian seal clade. At minimum, select synapomorphies for the Mesotaria/Sarcodectes + ?*Monotherium* + Monachini clade and Mesotaria/Sarcodectes + ?*Monotherium* clade should be included in the main text, since these are novel clades (if memory serves).

7) It hasn’t really been discussed much in the pinniped literature, but Van Beneden did the same thing with phocids that he did for whales: assembling chimaeric assemblages and not naming lectotypes, causing taxonomic paralysis and/or confusion for balaenopterid whales, cetotheriids, cetotheres sensu lato, and phocids for over a century. A bit of this, using *Herpetocetus scaldiensis*, is laid out by Demere et al. (2005) and El Adli et al. (2012). A brief discussion of this practice – and the apparent and unfortunate continuation of it by Koretsky et al. in the past 20 years – is warranted, particularly now that the Koretsky et al. “analysis” of “ecomorphotypes” has finally been “published” (Koretsky et al. 2020, First description of ecomorphotypes in seal subfamilies, *International Journal of Zoology and Animal Biology*). Nobody has said it yet, but Koretsky’s methods are essentially the same as Van Beneden’s imaginary seal constructs, just with a fancy name attached. This is a suggestion, but such an indictment is far from unwarranted.

8) On line 585-585 the authors cast doubt on the referral of an inominine from Spain to *Homiphoca* “given the various apparent morphotypes of *Homiphoca* from Langebaanweg...”, but on line 80-81 the authors state that “an initial study indicated there may be multiple species of seal...follow up work failed to support this...” and “as a result we have coded both crania and postcrania referred to *Homiphoca capensis* from Langebaanweg.” Is this a double standard being applied or a vestige of an earlier draft? If Rahmat et al. were right, and the Spanish specimen is *Homiphoca*, then credit is due.

9) Please include humeral features in the diagnosis. Diagnoses can be constructed at a hypodigm framework and should not be limited strictly to the holotype.

10) Lines 759-763: There are two issues here needing attendance. 1) Dewaele et al. 2018 reported a second large monachine seal from the Pliocene of Belgium, about the size of “*Callophoca*”. 2) “Whether this represents *Sarcodectes*...or if some of the Lee Creek Mine material is truly synonymous...” Shouldn’t this question be answered in this paper? As written, the authors seemingly leave the door open to *Callophoca obscura* possibly being diagnoseable after all.

11) Why do your results differ in some ways from the Churchill et al. results?

12) I’d like to see a couple of additions to the figures – a complete image of the cranium in lateral view, maybe mirroring the right side of the rostrum with the left braincase/zygoma; a line drawing reconstruction in dorsal/lateral/ventral would be an additional nice touch. For figure 7, enlarged views of all the teeth could easily be added onto the right side of the figure, blown up to 4x or so the size in the existing figures. As a matter of fact, there aren’t many informative photos of the distinctive teeth (that so much of the text is dedicated to describing and interpreting)

provided in the manuscript. The Western North Atlantic maps in Figure 13 are so zoomed in and cropped that it's difficult to properly interpret them – and those weird blobs for the fossil distribution – why not just do “x marks the spot”?

Minor comments

81: delete “work” or “study”

105: What exactly are the preferred methods? Which skull measurements do they use? It's a little ambiguous as written.

187: “inserted between the frontal”... frontals are being referred to collectively here (between) and should be plural

193: please quantify what low in height means

197: please quantify what large means

211: please include the collector's name here; ditto on line 217 for the referred specimen.

226: What member of the Yorktown Formation are these from?

228: Since that text is buried in a supplementary info document, you should probably just go ahead and cite the original sources here.

296: delete “is” after relatively

298: please quantify orbit size, perhaps as a % of skull width, and maybe present some comparisons with other phocids

305: please clarify “post auricular surface”. Are you talking about the squamosal fossa? That's generally what I've heard it termed.

357: Could you please quantify the angle that the tooththrows diverge at?

361: carina should be pluralized as carinae.

367: shelf on the cingulum?

370: “separated from PC4.”

373: ‘distinct’ may be overused in this text.

391: “underdeveloped” is certainly overused in this text and is not informative; better to say “fossae are shallow.”

510: underdeveloped again

511: “is shared with Hydrurga”

523: Based on which taxa from the Yorktown Formation?

523: Please clarify ‘larger estimated range of total body lengths’ – mean or max/min?

537: “the phocins are resolved” How are they resolved? This is pretty vague. They're not the focus of the current work, so you can get away with “resolved similar to analysis X or Y...”

592: As written, it sounds tentative and that you're about to lead into a discussion of comparisons, but that's already taken place; the next sentence goes into the infamous Oligocene specimens from Charleston. I suggest rewriting it to reiterate your earlier conclusions.

596: It would be good to make some reference to "Terranectes", which is certifiably a chimaera, but parts thereof may still represent some kind of a monachine.

624: Enlargement of the posterior oral cavity, and a shorter, wider rostrum/palate are generally associated with suction feeding (e.g. Werth, 2006, J. Mammalogy; Boessenecker et al., 2017: Proc B), so I am confused why this is evidence for raptorial feeding. It's also not really discussed again. Suggest clarifying or deleting.

648: Does *Neophoca* have sharp carinae? My impression is "not really".

654: There are also small dolphins present, like cf. *Pontoporia* and *Auroracetus*.

663: suggest 'extinct' rather than 'past'

670: This is somewhat unfair; Koretsky and Holec named the clade (we didn't, and our 2018 review paper is not an in depth taxonomic/phylogenetic evaluation), and they even performed a fairly bare-bones analysis. Much of Koretsky's work deserves skepticism but credit is needed where it is due, and at least a couple of analyses have confirmed her result. Perhaps frame it like "initial rudimentary phylogenetic hypotheses supported a stem-phocid position for *Devinophoca* (Koretsky and Holec, 2002), though later recovered as a stem phocine (Dewaele et al. 2017). We're all flattered of course but I think our phylogeny is based off of Fulton and Strobeck (2010), which has its own problems, and ours was a composite phylogeny (like those of Demere et al. 2003).

683: Use something else other than "a lot"

707-708: I think it's usually just cited as Muizon rather than de Muizon in text

713: our (Boessenecker and Churchill's) 2016 paper on the Waipunga seal from NZ also cast doubt on *Miroungin* affinities of *Callophoca* aside from "it's a big seal".

718: "recently been put forward" a bit unclear/awkward

721: Typo - *Sarcodectes*

723: "put forward" again

723: what types of analyses? As written it assumes the reader is already familiar with these analyses

744-749: add some comparison measurements here

750: size of the fauna or body sizes within the fauna?

768-769: This is misleading, as the skull of *Mesotaria*/*Sarcodectes*/*Callophoca* has been coded in prior analyses. As written, the reader could mistake this for meaning that phylogenetic analysis of this specimen was attempted here for the first time - which is not true.

Figure 3 caption: 'ear region' could probably be replaced with squamosal.

Figure 13: please list the taxa, either on the image, or in the caption, and maybe provide the individual data points on the graph on the left, above the bars, for clarity.

Reviewer: 2

Comments to the Author(s)

Very clear manuscript. In the abstract, please constrain the timing of body size increase. I suggest including the body-size analysis into the title. It is a more important contribution of the study compared to the report of unusual dentition. Besides, "unusual dentition" alone is not informative. I also suggest changing the title of the section "Functional implications" to something more informative and specific.

Questions I suggest addressing:

1. The authors added new states to several characters, which had an impact on the phylogeny. Have the authors considered whether these multistate characters can be ordered? If the states form a morphocline and ordered that may impact their analysis. What is the reason for unordering all characters? Multistate characters that can be organized to form a morphocline should be ordered otherwise why forming a multistate character in the first place in those cases?
2. To which extent previous alternative phylogenies would change the overall pattern found by the body size analysis of the current study?

Author's Response to Decision Letter for (RSOS-200796.R0)

See Appendix A.

RSOS-201591.R0

Review form: Reviewer 1 (Robert Boessenecker)

Is the manuscript scientifically sound in its present form?

No

Are the interpretations and conclusions justified by the results?

No

Is the language acceptable?

Yes

Do you have any ethical concerns with this paper?

No

Have you any concerns about statistical analyses in this paper?

No

Recommendation?

Major revision is needed (please make suggestions in comments)

Comments to the Author(s)

In general I am extremely satisfied and impressed with the minor and moderate corrections, and were it up to these changes alone, I would recommend immediate acceptance. However, I have not been convinced beyond a reasonable doubt by the new text and figures addressing my major comment, regarding whether or not this seal is conspecific or congeneric with *Mesotaria ambigua* of Van Beneden. From my perspective, the differences between USNM 534034 and the *Mesotaria*

ambigua holotype seem quite minor and quite possibly within the range of intraspecific variation, let alone the possibility of variation from sexual dimorphism. My background is chiefly within “otarioids” – but I will say that, using an example of a well-sampled walrus from the west coast, there is more variation within the humeri of *Valenictus chulavistensis* than between these (to be specific - shape of the distal trochlea, entepicondyles, and deltopectoral crest). I understand that the authors at this point are perhaps wedded to the opportunity to name a new binomial, but I am not convinced that they have made a strong case that this is distinct from *Mesotaria ambigua*. The efforts to distinguish these humeri come across as hair-splitting. Given the similarity, shouldn't these be congeneric, at a minimum? I think, at present, the only way to demonstrate that this is truly a different species (or genus) would be to conduct a morphometric study of the humeri and compare it with the range of variation for other phocids. This sounds like a lot of work - but is probably necessary to demonstrate that there's more to this than qualitative hair-splitting.

This opens a different can of worms: what to do with old holotypes erected during a time of different standards? Even going back to establishing *Callophoca obscura* as a nomen dubium – do we do the same for many cetaceans? *Zarhachis* was stabilized by Kellogg – the type specimen is a vertebra – but the taxonomy is now considered stable around a skull. Do we get rid of *Zarhachis*? *Delphinodon dividum* is not the same morphotype as the type species, *Delphinodon mento*, likely a *Hadrodelphis*-like dolphin; what do we do there? What about *Zygorhiza kochii* – the holotype is pretty scrappy, and Uhen proposed designating a neotype – which Gingerich (2015) railed against for various reasons, and ultimately the ICZN declined to allow neotype designation. I'm not certain which philosophy I agree with, though I have generally opted to follow Kellogg's lead and stabilize old names with the referral of better specimens. I suggest touching up on Romer (1968: Notes and Comments on Vertebrate Paleontology – chapter 1) and mulling it over a bit.

Sorry to be a bit of a stick in the mud on this issue.

Cheers, Robert Boessenecker, Ph.D.

Review form: Reviewer 2 (Márton Rabi)

Is the manuscript scientifically sound in its present form?

Yes

Are the interpretations and conclusions justified by the results?

Yes

Is the language acceptable?

Yes

Do you have any ethical concerns with this paper?

No

Have you any concerns about statistical analyses in this paper?

No

Recommendation?

Accept as is

Comments to the Author(s)

Thank you for your responses, my questions were mostly answered.

Decision letter (RSOS-201591.R0)

Dear Mr Rule

The Editors assigned to your paper RSOS-201591 "A new large-bodied Pliocene seal with unusual cutting teeth" have now received comments from reviewers and would like you to revise the paper in accordance with the reviewer comments and any comments from the Editors. Please note this decision does not guarantee eventual acceptance.

Please submit your revised manuscript and required files (see below) no later than 21 days from today's (ie 29-Sep-2020) date. Note: the ScholarOne system will 'lock' if submission of the revision is attempted 21 or more days after the deadline. If you do not think you will be able to meet this deadline please contact the editorial office immediately.

on behalf of Dr Denise Greig (Associate Editor) and Kevin Padian (Subject Editor)
openscience@royalsociety.org

Associate Editor Comments to Author (Dr Denise Greig):

Thank you for responding so thoroughly to the previous reviewer comments. Dr Boessenecker was likewise impressed, but has one remaining concern regarding whether *Sarcodectes magnus* can be described as separate from *Mesotaria ambigua*, and whether we need a more in depth analysis of humeri across the pinnipeds.

From the journal's standpoint, a new taxon will need enough synapomorphy(ies) that others can use to recognize it when they see it in another specimen. If not, then you should consider the level of taxon to which the animal, with the diagnostic features that it has, should be assigned.

Reviewer comments to Author:

Reviewer: 1

Comments to the Author(s)

In general I am extremely satisfied and impressed with the minor and moderate corrections, and were it up to these changes alone, I would recommend immediate acceptance. However, I have not been convinced beyond a reasonable doubt by the new text and figures addressing my major comment, regarding whether or not this seal is conspecific or congeneric with *Mesotaria ambigua* of Van Beneden. From my perspective, the differences between USNM 534034 and the *Mesotaria ambigua* holotype seem quite minor and quite possibly within the range of intraspecific variation, let alone the possibility of variation from sexual dimorphism. My background is chiefly within "otarioids" – but I will say that, using an example of a well-sampled walrus from the west coast, there is more variation within the humeri of *Valenictus chulavistensis* than between these (to be specific - shape of the distal trochlea, entepicondyles, and deltopectoral crest). I understand that the authors at this point are perhaps wedded to the opportunity to name a new binomial, but I am not convinced that they have made a strong case that this is distinct from *Mesotaria ambigua*. The efforts to distinguish these humeri come across as hair-splitting. Given the similarity, shouldn't these be congeneric, at a minimum? I think, at present, the only way to demonstrate that this is truly a different species (or genus) would be to conduct a morphometric study of the humeri and compare it with the range of variation for other phocids. This sounds like a lot of work - but is probably necessary to demonstrate that there's more to this than qualitative hair-splitting.

This opens a different can of worms: what to do with old holotypes erected during a time of different standards? Even going back to establishing *Callophoca obscura* as a nomen dubium – do we do the same for many cetaceans? *Zarhachis* was stabilized by Kellogg – the type specimen is a vertebra – but the taxonomy is now considered stable around a skull. Do we get rid of *Zarhachis*? *Delphinodon dividum* is not the same morphotype as the type species, *Delphinodon mento*, likely a *Hadrodelphis*-like dolphin; what do we do there? What about *Zygorhiza kochii* – the holotype is pretty scrappy, and Uhen proposed designating a neotype – which Gingerich (2015) railed against for various reasons, and ultimately the ICZN declined to allow neotype designation. I'm not certain which philosophy I agree with, though I have generally opted to follow Kellogg's lead and stabilize old names with the referral of better specimens. I suggest touching up on Romer (1968: Notes and Comments on Vertebrate Paleontology – chapter 1) and mulling it over a bit.

Sorry to be a bit of a stick in the mud on this issue.

Cheers, Robert Boessenecker, Ph.D.

Reviewer: 2

Comments to the Author(s)

Thank you for your responses, my questions were mostly answered.

===PREPARING YOUR MANUSCRIPT===

- one version identifying all the changes that have been made (for instance, in coloured highlight, in bold text, or tracked changes);
- a 'clean'

version of the new manuscript that incorporates the changes made, but does not highlight them. This version will be used for typesetting if your manuscript is accepted.

===PREPARING YOUR REVISION IN SCHOLARONE===

- Ensure that your data access statement meets the requirements at <https://royalsociety.org/journals/authors/author-guidelines/#data>. You should ensure that you cite the dataset in your reference list. If you have deposited data etc in the Dryad repository, please include both the 'For publication' link and 'For review' link at this stage.
- If you are requesting an article processing charge waiver, you must select the relevant waiver option (if requesting a discretionary waiver, the form should have been uploaded at Step 3 'File upload' above).
- If you have uploaded ESM files, please ensure you follow the guidance at <https://royalsociety.org/journals/authors/author-guidelines/#supplementary-material> to include a suitable title and informative caption. An example of appropriate titling and captioning may be found at https://figshare.com/articles/Table_S2_from_Is_there_a_trade-off_between_peak_performance_and_performance_breadth_across_temperatures_for_aerobic_scope_in_teleost_fishes_/3843624.

Author's Response to Decision Letter for (RSOS-201591.R0)

See Appendix B.

Decision letter (RSOS-201591.R1)

Dear Mr Rule,

It is a pleasure to accept your manuscript entitled "A new large-bodied Pliocene seal with unusual cutting teeth" in its current form for publication in Royal Society Open Science. The comments of the reviewer(s) who reviewed your manuscript are included at the foot of this letter.

on behalf of Dr Denise Greig (Associate Editor) and Kevin Padian (Subject Editor)
openscience@royalsociety.org

Appendix A

Dear editors (Dr Denise Greig, Prof. Kevin Padian) and Reviewers,

We have attached here our responses to the reviewer's comments for our manuscript (**A new large-bodied Pliocene seal with unusual cutting teeth**), which were very helpful in strengthening the paper. The comments regarding the taxonomic issue of *Mesotaria* were extremely helpful, as this was an essential issue to resolve. We have responded to individual comments below, most for which we have made the requested changes. For a few comments, suggestions were made which were beyond the practical scope of the paper. We have responded in detail to these suggestions, but reiterate that the primary focus of the paper is the taxonomic problems regarding large fossil monachines, and how solving these issues changes what we know about phylogeny and body size in ancient Monachinae.

Reviewers' Comments to Author:

Reviewer: 1

Comments to the Author(s)

This manuscript is a competent addition to the literature on fossil seals, and provides a much needed, albeit limited, revision of the large bodied Pliocene seal *Callophoca obscura* from the North Atlantic. *Callophoca* was originally named based off of a fairly eroded but large humerus missing most of the anatomically informative proximal and distal ends, collected from the Pliocene of Belgium. Van Beneden assigned a "type series" of non-associated fossils to *Callophoca*, and a lectotype humerus was designated by Koretsky and Ray (2008). This latter study was a bit of a dog's breakfast, repeating many of the same mistakes as Van Beneden – assigning nonassociated fossils from the Lee Creek Mine in North Carolina to a morphotype based upon a preconceived notion of what the different parts of the seal "should" look like, versus emphasizing consideration of overlapping parts only. One of the specimens, referred originally on size, is a partial skull, here designated as the holotype of *Sarcodectes magnus*. This new species is competently described with particular attention towards interpreting its feeding ecology and the evolution of body size in monachine seals.

However, there is a serious taxonomic problem buried within this. The lectotype specimen of *Callophoca obscura*, IRSNB 1198, is not informative, and the authors are indeed well-supported in their bid to designate it a nomen dubium. However, the path forward to name a new species or genus is far from clear, and aside from tossing out *Callophoca*, little attention is paid in the manuscript to other Belgian fossil phocids, which are sort of sidestepped. Koretsky and Ray (2008) also designated a lectotype for *Mesotaria ambigua*, IRSNB 1156, which is a nearly complete humerus missing only the tip of the lesser tuberosity. This humerus, figured quite nicely by PJ Van Beneden 1877: plate 9.9-9.11, is nearly identical to the humerus of USNM 534034, referred in this manuscript to *Sarcodectes magnus* and linked to the proposed holotype cranium by a squamosal. The *Mesotaria ambigua* lectotype, unlike that of *Callophoca obscura*, is diagnostic and the same arguments for a nomen dubium cannot be made. Furthermore, Koretsky and Ray (2008) declared *Mesotaria ambigua* a synonym of *Callophoca obscura* and referred the lectotype humerus to *Callophoca*, so there is existing published opinion that the *Mesotaria ambigua* lectotype humerus and the proposed *Sarcodectes magnus* holotype and referred specimens are part of the same hypodigm. A second even better preserved humerus exists, figured by Koretsky and Ray (2008: figure 24a, 25a, 26a, 27a) and referred to *Mesotaria* by Van Beneden 1876 but not figured. In my opinion, caution is warranted and at present I think a case for a new genus/species is not strong, particularly given that no

mention of *Mesotaria ambigua* is made in the present manuscript. Owing to the similarity of the *Mesotaria ambigua* lectotype with USNM 534034, this taxon, owing to taxonomic priority, is probably better referred to as *Mesotaria ambigua* or perhaps a new species of *Mesotaria* (*Mesotaria sarcodectes*, perhaps? That way you could keep that killer name). Detailed comparisons have yet to be made and are a starting point for addressing this issue.

Otherwise, I am very impressed with the current manuscript and I commend the level of detail, and have a number of moderate and minor suggestions that could improve the long term utility of the work.

I commend Mr. Rule on his excellent work and have no concerns that he and the rest of the team will be able to take an already great paper and make it into a fantastic one. Kind regards, Robert Boessenecker

Moderate comments

1) As mentioned above, Belgian specimens/taxa aside from *Callophoca obscura* are sort of side-stepped. A further issue is that USNM 534034 is not the only large monachine humerus from the Lee Creek Mine, and there are others figured by Koretsky and Ray (2008). Are these, or are they not, the same taxon? For example: USNM 186944, 263656. Further to the point, the authors claim that *Mesotarial/Sarcodectes* is known only from the Western North Atlantic, but have not bothered making any comparisons with Belgian specimens of “*Callophoca*” referred/figured by Koretsky and Ray that happen to be quite well preserved (e.g. IRSNB 301).

- We agree that *Mesotaria* is critical to address in order for a solid taxonomic argument to be made, and we have expanded our text to reflect this. Our evaluation of the *Mesotaria* lectotype, however, does not lead us to revise our treatment of USNM PAL 534034 as *Sarcodectes magnus*. This is based on clear differences in the distal humeral morphology, which are now outlined in the diagnosis [lines 191–195], comparisons section [lines 503–516], and the discussion subheading “Implications for monachine taxonomy” [lines 598–603]. We have also added two new Figures (14 and 15) in order to demonstrate the differences between *Sarcodectes*, *Mesotaria*, and *Auroraphoca* (the three large monachines from the North Atlantic region). These morphological differences are preserved in both the *Mesotaria* lectotype, and USNM PAL 534034, but not in the old “*Callophoca*” lectotype. It is on this basis that we retain our recommendation that “*Callophoca obscura*” is a *nomen dubium*. The retention of the name *Sarcodectes magnus* is based on type specimen comparisons; however, the reviewer is correct that other previously referred specimens need to be considered. To address both this and moderate comment 10, we have added additional text in “Implications for monachine taxonomy” [lines 598–603], and have added a table to address which specimens should be referred to *Sarcodectes magnus*, *Auroraphoca atlantica*, or *Mesotaria ambigua* [Table 2].

2) Please consider including the holotype skull measurements as a table within the main text.

- We have moved both the measurements for the holotype skull and the measurements for the paratype specimens into the main text, in order to keep them in the same place. We have made changes to line 78 to reflect this.

3) Why not compare the range of body sizes, at a faunal level, with those from the North Pacific? Also, regarding pinniped assemblages as a whole, it may be worth mentioning that there is a much larger bodied walrus (*Otocetus emmonsii*) in the North Atlantic – a bit odd to leave out of a faunal discussion. Also, as regards the extant Antarctic assemblage, some explanation/hypotheses for why there are larger body sizes across the board would be welcome – higher primary productivity, colder water temperatures, perhaps? Some combination?

- We agree that some expansion to other pinniped groups would be useful for the Body Size discussion section, and so we have added some text to address this [lines 791–792]. This includes reference to *Otocetus* [lines 781–783]. However, for the broad-scale comparisons in Figure 18 and the text, we have chosen to mostly stick to phocids, as assessing pinnipeds as a whole is beyond the scope of this study. We (the authors) maintain that the focus of the paper is the taxonomic reassessment of *Callophoca/Sarcodectes*, and that any broader scale patterns (phylogeny, body size) are initial analyses and discussions based on these reassessments. A lot more work on pinniped taxonomy as a whole need to be done before site specific, broad scale comparisons can be made.

4) It would be wise to include a figure of the *Callophoca obscura* lectotype humerus IRSNB 1198, in the interest of transparency.

- We agree, and have included a figure of the lectotype as Figure 1.

5) In various places ‘large’ and ‘small’ are used in the absence of quantification; please include brief quantifiers in the text (e.g. small, <10% of bizygomatic width).

- We have addressed these instances in some cases where it is appropriate. However, in a few cases, depending on the context, we have judged that quantification is not as important as the relative size of morphology in a qualitative aspect. This is mostly to avoid choosing an arbitrary quantitative “cut-off” value, which would require a large (and separate) quantitative comparative analysis across Phocidae to justify.

6) Some more synapomorphies should be listed in the main text, as written the inclusion of synapomorphies is a bit lopsided. Better yet, including them on an annotated cladogram would really help – just within the Monachinae, anyway, or maybe even just the Monachini + *Mesotaria/Sarcodectes* + Peruvian seal clade. At minimum, select synapomorphies for the *Mesotaria/Sarcodectes* + ?*Monotherium* + Monachini clade and *Mesotaria/Sarcodectes* + ?*Monotherium* clade should be included in the main text, since these are novel clades (if memory serves).

- We agree that the synapomorphy results could be more refined, and have made some modifications to this section of the main text [lines 549–555], as well as a more extensive Supplemental Table 6 (listing the clade specific synapomorphies). However, due to the novel clades either having relatively few or no unequivocal synapomorphies, we have decided not to include them in an annotated cladogram. The lack of unequivocal synapomorphies is the reason why the synapomorphy results reported are so limited, as discussion of equivocal synapomorphies would not be beneficial to the interpretation of the results.

7) It hasn’t really been discussed much in the pinniped literature, but Van Beneden did the same thing with phocids that he did for whales: assembling chimaeric assemblages and not naming lectotypes, causing taxonomic paralysis and/or confusion for balaenopterid whales, cetotheriids, cetotheres sensu lato, and phocids for over a century. A bit of this, using *Herpetocetus scaldiensis*, is laid out by Demere et al. (2005) and El Adli et al. (2012). A brief discussion of this practice – and the

apparent and unfortunate continuation of it by Koretsky et al. in the past 20 years – is warranted, particularly now that the Koretsky et al. “analysis” of “ecomorphotypes” has finally been “published” (Koretsky et al. 2020, First description of ecomorphotypes in seal subfamilies, International Journal of Zoology and Animal Biology). Nobody has said it yet, but Koretsky’s methods are essentially the same as Van Beneden’s imaginary seal constructs, just with a fancy name attached. This is a suggestion, but such an indictment is far from unwarranted.

- We agree that this needs to be highlighted in the literature, and so have added this discussion to the subheading “Implications for monachine taxonomy” [lines 566–587].

8) On line 585-585 the authors cast doubt on the referral of an inominat from Spain to *Homiphoca* “given the various apparent morphotypes of *Homiphoca* from Langebaanweg...”, but on line 80-81 the authors state that “an initial study indicated there may be multiple species of seal...follow up work failed to support this...” and “as a result we have coded both crania and postcrania referred to *Homiphoca capensis* from langebaanweg.” Is this a double standard being applied or a vestige of an earlier draft? If Rahmat et al. were right, and the Spanish specimen is *Homiphoca*, then credit is due.

- We agree that this logic is circular, and is definitely a vestige from various drafts. We have instead replaced the argument on why there are concerns with the referral of the inominat from Spain to *Homiphoca* [the fossil is rather fragmentary, lines 611–612]. We have also made some alterations to the methods section to make this treatment more clear [lines 61–62].

9) Please include humeral features in the diagnosis. Diagnoses can be constructed at a hypodigm framework and should not be limited strictly to the holotype.

- There is already one unique humeral feature in the diagnosis (rounded, thick medial epicondyle with a proximally located fossa on the anterior end). We have added additional features of the humerus to the diagnosis, on the basis that only this combination of humeral morphology is diagnostic. We have also modified the existing humeral feature so it is more specific [lines 191–195].

10) Lines 759-763: There are two issues here needing attendance. 1) Dewaele et al. 2018 reported a second large monachine seal from the Pliocene of Belgium, about the size of “*Callophoca*”. 2) “Whether this represents *Sarcodectes*...or if some of the Lee Creek Mine material is truly synonymous...” Shouldn’t this question be answered in this paper? As written, the authors seemingly leave the door open to *Callophoca obscura* possibly being diagnoseable after all.

- We agree this can be resolved within this paper. Using the humeral features in the diagnosis, we have added discussion addressing these specimens in the discussion subheading “Implications for monachine taxonomy” [lines 598–603], and altered the discussion on lines 798–802 addressing body size to reflect these changes.

11) Why do your results differ in some ways from the Churchill et al. results?

- We have altered the text in the “body size evolution” subheading to further explore both the methodological differences, as well as a more thorough comparison of the results [lines 761–780]. We have also updated the results for the ancestral state estimation [lines 556–560] to include more details to allow for meaningful comparisons with Churchill *et al.* 2015.

12) I’d like to see a couple of additions to the figures – a complete image of the cranium in lateral view, maybe mirroring the right side of the rostrum with the left

braincase/zygoma; a line drawing reconstruction in dorsal/latera/ventral would be an additional nice touch. For figure 7, enlarged views of all the teeth could easily be added onto the right side of the figure, blown up to 4x or so the size in the existing figures. As a matter of fact, there aren't many informative photos of the distinctive teeth (that so much of the text is dedicated to describing and interpreting) provided in the manuscript. The Western North Atlantic maps in Figure 13 are so zoomed in and cropped that it's difficult to properly interpret them – and those weird blobs for the fossil distribution – why not just do “x marks the spot”?

- We have addressed this by adding to figures to the main text: 1. A line drawing reconstruction, that has the full lateral view as has been requested. 2. A separate figure with close ups of the dentition in various views. We have also added to the Supplemental Information lateral and oblique views of the reconstructed skull.

We have also made the alterations to figures 7 and 13 (old numbering) as requested.

Minor comments

81: delete “work” or “study”

- Corrected, “study” deleted.

105: What exactly are the preferred methods? Which skull measurements do they use? It's a little ambiguous as written.

- We have deleted “methods” as the equations that follow are the described methods. We've added in parentheses that both equations use multiple skull metrics.

187: “inserted between the frontal” ... frontals are being referred to collectively here (between) and should be plural

- This has been corrected.

193: please quantify what low in height means

- This sentence has been altered to specify “less than twice the height of the accessory cusps”.

197: please quantify what large means

- We have chosen not to quantify this term, as the important observation here for the purposes of the diagnosis section is that the PC4–PC5 diastema is larger relative to all the other diastemas in the postcanine toothrow, rather than the degree to which it is larger.

211: please include the collector's name here; ditto on line 217 for the referred specimen.

- Collectors names have been added here.

226: What member of the Yorktown Formation are these from?

- Information of the specific member from the Yorktown Formation these fossils are from is not associated with these specimens. It is for this reason that these specimens are treated as being from the broader Yorktown Formation throughout the manuscript.

228: Since that text is buried in a supplementary info document, you should probably just go ahead and cite the original sources here.

- Agreed, we have added these citations.

296: delete “is” after relatively

- Deleted.

298: please quantify orbit size, perhaps as a % of skull width, and maybe present some comparisons with other phocids

- We have presented the percentage of skull width in the description, and also added some text to line 469 of the comparison section to address that this is presented as a percentage of skull width.

305: please clarify “post auricular surface”. Are you talking about the squamosal fossa? That’s generally what I’ve heard it termed.

- This is what we mean specifically, and we have modified the text to reflect this.

357: Could you please quantify the angle that the toothrows diverge at?

- This has been added as an angle from the midline.

361: carina should be pluralized as carinae.

- This has been added to the text.

367: shelf on the cingulum?

- This is not referring to the cingulum, as the cingulum is a development on the outside border of the tooth. Rather, this is referring to the surface of the crown between the main cusp/cusplets and the cingulum. We have removed “facing: to make this more clear.

370: “separated from PC4.”

- We agree that this is simpler, and have made this change.

373: ‘distinct’ may be overused in this text.

- This is only the second use of “distinct” in this paragraph, and so we have chosen to keep it.

391: “underdeveloped” is certainly overused in this text and is not informative; better to say “fossae are shallow.”

- We agree, and have replaced “underdeveloped” with “shallow”.

510: underdeveloped again

- This has been altered to clarify that the lateral epicondyle of the humerus is not prominent.

511: “is shared with *Hydrurga*”

- We have made this change.

523: Based on which taxa from the Yorktown Formation?

- This is defined in the Methods section for Body Size.

523: Please clarify ‘larger estimated range of total body lengths’ – mean or max/min?

- We have clarified with “min–max” in parentheses.

537: “the phocins are resolved” How are they resolved? This is pretty vague. They’re not the focus of the current work, so you can get away with “resolved similar to analysis X or Y...”

- We can see the confusion of the sentence as is; we mean to say that the relationships are resolved in a fully bifurcating clade (not a polytomy). We have clarified the meaning in text. We have not included a comparison to other studies as this text is part of the results section.

592: As written, it sounds tentative and that you're about to lead into a discussion of comparisons, but that's already taken place; the next sentence goes into the infamous Oligocene specimens from Charleston. I suggest rewriting it to reiterate your earlier conclusions.

- We have reworded this sentence and relocated it to the beginning of the paragraph for clarity.

596: It would be good to make some reference to "Terranectes", which is certifiably a chimaera, but parts thereof may still represent some kind of a monachine.

- We have added a sentence addressing this Genus, and Dewaele *et al.*'s suggestion it be considered a *nomen dubium*.

624: Enlargement of the posterior oral cavity, and a shorter, wider rostrum/palate are generally associated with suction feeding (e.g. Werth, 2006, J. Mammalogy; Boessenecker et al., 2017: Proc B), so I am confused why this is evidence for raptorial feeding. It's also not really discussed again. Suggest clarifying or deleting.

- This was originally included as large posterior oral cavities are a feature of terrestrial carnivorans with high bite forces. However, as this morphology conflates with suction feeding, and is not brought up again, we have chosen to delete this sentence.

648: Does *Neophoca* have sharp carinae? My impression is "not really".

- *Neophoca* does have some sharp carinae on the main cusps of their teeth; this is visible in a figure of the paper we cite in this line regarding *Neophoca* (Hocking *et al.* 2017. Chew, shake, and tear: Prey processing in Australian sea lions (*Neophoca cinerea*).

654: There are also small dolphins present, like cf. *Pontoporia* and *Auroracetus*.

- We have added dolphins to this list, along with references to these two genera.

663: suggest 'extinct' rather than 'past'

- We agree, and have switched to "extinct".

670: This is somewhat unfair; Koretsky and Holec named the clade (we didn't, and our 2018 review paper is not an in depth taxonomic/phylogenetic evaluation), and they even performed a fairly bare-bones analysis. Much of Koretsky's work deserves skepticism but credit is needed where it is due, and at least a couple of analyses have confirmed her result. Perhaps frame it like "initial rudimentary phylogenetic hypotheses supported a stem-phocid position for *Devinophoca* (Koretsky and Holec, 2002), though later recovered as a stem phocine (Dewaele et al. 2017). We're all flattered of course but I think our phylogeny is based off of Fulton and Strobeck (2010), which has its own problems, and ours was a composite phylogeny (like those of Demere et al. 2003).

- We agree that this comparison isn't appropriate. We have removed the citation from this sentence to the Berta *et al* 2018 review, and retained the citations to the papers by Koretsky and colleagues.

683: Use something else other than "a lot"

- We have replaced "a lot" with "substantially". We have also made another refinement to this sentence so it is more accurate [line 711].

707-708: I think it's usually just cited as Muizon rather than de Muizon in text

- Corrected in the text by removing "de".

713: our (Boessenecker and Churchill's) 2016 paper on the Waipunga seal from NZ also cast doubt on Miroungin affinities of *Callophoca* aside from "it's a big seal".

- While this is true, we have stuck to discussion of phylogenetic analyses including the holotype specimen in this section of the text. As such, we have chosen not to cite this paper here (it is cited later), so as to avoid discussion of the relationships of "*Callophoca obscura*" in general, which at this stage of the text has already been stated to be a *nomen dubium*.

718: "recently been put forward" a bit unclear/awkward

- We have replaced this with "proposed".

721: Typo – *Sacrodectes*

- Corrected.

723: "put forward" again

- We have replaced this with "that have previously been suggested".

723: what types of analyses? As written it assumes the reader is already familiar with these analyses

- We have specified "phylogenetic and biogeographic analyses".

744-749: add some comparison measurements here

- We prefer not to add specific measurements to the text of the discussion, as the figure exists to improve the readability of the manuscript. As a compromise, we have added the ranges to the Results section 521–526, where their inclusion is appropriate. This way, the values are reported in the results and the Supplemental Information, and do not need to be repeated in the discussion.

750: size of the fauna or body sizes within the fauna?

- We have clarified that it is within the fauna.

768-769: This is misleading, as the skull of *Mesotaria/Sarcodectes/Callophoca* has been coded in prior analyses. As written, the reader could mistake this for meaning that phylogenetic analysis of this specimen was attempted here for the first time – which is not true.

- We have altered this sentence for clarity. However, we specify that it enables the "reliable" inclusion of this taxon; all other inclusions of this specimen have been as part of chimeras, which is not particularly useful. This precise point was brought up in Berta *et al.* 2015, in their discussion of the treatment of "*Callophoca*" in their phylogenetic analysis. It is the taxonomic chimeric problem in phylogenetic analyses and taxonomy that led to this study in the first place, so we very much want to reiterate this in the concluding paragraph. We hope that our alteration of the sentence means this can no longer be read as misleading.

Figure 3 caption: 'ear region' could probably be replaced with squamosal.

- We agree, and have made this change.

Figure 13: please list the taxa, either on the image, or in the caption, and maybe provide the

individual data points on the graph on the left, above the bars, for clarity.

- We have listed the taxa in the figure caption, and added the data points to the graph.

Reviewer: 2

Comments to the Author(s)

Very clear manuscript.

In the abstract, please constrain the timing of body size increase.

- The timing of body size increase would be interesting to look at. However, because the nodes were dated by the R package “strap” (which only uses the stratigraphic dates of the tip, and are arbitrarily calibrated by the user), and because the heatmap for the ancestral state estimation of body size is not informed by branch lengths on the tree, it is inappropriate to make any inferences on the timing of the body size increase.

I suggest including the body-size analysis into the title. It is a more important contribution of the study compared to the report of unusual dentition. Besides, "unusual dentition" alone is not informative.

- We have decided to keep the title as is, as it currently reflects that the seal is “large”, and the dentition/feeding implications (which contribute to more of the discussion than the body size analysis). This will also mean the title is more concise, and better reflect the focus on the newly described taxon.

I also suggest changing the title of the section "Functional implications" to something more informative and specific.

- We have changed the title of this section to “Functional implications for feeding behaviour”.

Questions I suggest addressing:

1. The authors added new states to several characters, which had an impact on the phylogeny. Have the authors considered whether these multistate characters can be ordered? If the states form a morphocline and ordered that may impact their analysis. What is the reason for unordering all characters? Multistate characters that can be organized to form a morphocline should be ordered otherwise why forming a multistate character in the first place in those cases?

- We have considered ordering the characters, but due to the small size of the phylogeny (in terms of extinct taxa included) we thought it safer to run the analysis without the assumption of ordering characters. While it is true that some of the new and modified characters would form a “morphocline”, they were not constructed for that purpose. Rather, it was because the existing morphological characters in the literature for Phocidae either did not sufficiently represent several areas of morphology (e.g. the ear region, some postcranial elements), or because the morphology in some taxa were not represented by existing states for morphological characters from previous analyses.

2. To which extent previous alternative phylogenies would change the overall pattern found by the body size analysis of the current study?

- Although we agree this would be interesting to look into, we believe this is beyond the scope of this paper describing a new Genus and species of phocid. We feel this question is not appropriate to explore in this paper for several reasons:

- 1) There are no equivalent phylogenies (in terms of number of taxa or included taxa) in the literature.
- 2) It will be difficult to make these comparisons regardless, due to the previous chimeric treatment of "*Callophoca obscura*".
- 3) The focus (and interest) of the inclusion of body size in this paper is *how* the description of this new taxon, and the declaration of "*Callophoca obscura*" as a *nomen dubium*, has altered what we know about the evolution of body size in Monachinae. We (the authors) do not view the body size analysis itself to be the central focus of the manuscript.

Appendix B

Associate Editor Comments to Author (Dr Denise Greig):

Thank you for responding so thoroughly to the previous reviewer comments. Dr Boessenecker was likewise impressed, but has one remaining concern regarding whether *Sarcodectes magnus* can be described as separate from *Mesotaria ambigua*, and whether we need a more in depth analysis of humeri across the pinnipeds. From the journal's standpoint, a new taxon will need enough synapomorphy(ies) that others can use to recognize it when they see it in another specimen. If not, then you should consider the level of taxon to which the animal, with the diagnostic features that it has, should be assigned.

We appreciate the overall positive feedback to our initial revisions and have outlined our reasoning in our direct response below. In sum, there are three active areas of concern: 1) the appropriate taxonomic allocation of the fossil specimens; 2) whether further analysis would positively support a particular taxonomic allocation; 3) whether our fossil specimens exhibit sufficient synapomorphies to warrant a new binomial.

Regarding the remaining concern by Dr. Boessenecker around our analysis and decision to not attribute specimens to *Mesotaria*, **we ultimately do not find an evidentiary basis to refer the described specimens to this genus.** We describe the material as a separate genus and species because of the current consensus in the phocid taxonomic literature; that phocid humeri are uninformative types, and therefore descriptions of, and referrals to, isolated humeri cannot be made. We understand that these opposing stances may represent differing philosophical approaches to taxonomy and are not reconcilable. But based on the published science and ICZN rules, our approach is rigorous, biologically conservative, and valid for the treatment of these fossil specimens.

In regards to whether a more in-depth (morphometric) analysis is needed, one has already been published. As we cite in our manuscript, Churchill and Uhen (2019) performed a morphometric study to test whether isolated phocid humeri were taxonomically informative. Their results (Fig 3, pg 218) demonstrate that morphometric methods poorly distinguish taxa even at the level of subfamily, and fail to differentiate taxa at lower levels. It also demonstrated that very few genera or species cluster in the morphospace, and extensive overlap was found between genera and species. In addition, a qualitative review of morphological characters of the humerus traditionally used in taxonomy of phocids found almost all either unreliable or only useful at distinguishing between subfamilies. The paper concluded that isolated humeri (and femora) are not useful type specimens, and should either be considered *nomen dubia* or treated with caution. Given the strength and results of this recent analysis, we have no basis for suggesting that another analysis will provide positive support on this issue as suggested.

The new taxon has several synapomorphies that meet the appropriate threshold for erecting a new binomial (as required by the journal and the ICZN). The synapomorphies, and the combination of other morphological characters outlined, are not present on any other species of phocid. As detailed in our revised manuscript, none of the following features are present in any other monachines: the distinct well developed intercuspid notches; the enlarged convex protuberance on the mastoid; the groove on the postglenoid process.

Because isolated humeri have been shown to be uninformative, referrals to humeri-based taxa should only be done if the specimen being referred is identical to the humerus-type. Because *Mesotaria* is found at a locality on the other side of the Atlantic, at a different point in time (separated by 0.5 million years), referral of humeri outside this spatial and temporal

region would have serious biological and biogeographical consequences. Considering the noted and published unreliability of humerus-based taxonomic attribution, and since the humerus of USNM PAL 534034 does have several distinct morphological differences to the *Mesotaria* lectotype, it cannot be reliably referred to that name.

Reviewer: 1

In general I am extremely satisfied and impressed with the minor and moderate corrections, and were it up to these changes alone, I would recommend immediate acceptance. However, I have not been convinced beyond a reasonable doubt by the new text and figures addressing my major comment, regarding whether or not this seal is conspecific or congeneric with *Mesotaria ambigua* of Van Beneden. From my perspective, the differences between USNM 534034 and the *Mesotaria ambigua* holotype seem quite minor and quite possibly within the range of intraspecific variation, let alone the possibility of variation from sexual dimorphism. My background is chiefly within “otarioids” – but I will say that, using an example of a well-sampled walrus from the west coast, there is more variation within the humeri of *Valenictus chulavistensis* than between these (to be specific - shape of the distal trochlea, entepicondyles, and deltopectoral crest). I understand that the authors at this point are perhaps wedded to the opportunity to name a new binomial, but I am not convinced that they have made a strong case that this is distinct from *Mesotaria ambigua*. The efforts to distinguish these humeri come across as hair-splitting. Given the similarity, shouldn't these be congeneric, at a minimum? I think, at present, the only way to demonstrate that this is truly a different species (or genus) would be to conduct a morphometric study of the humeri and compare it with the range of variation for other phocids. This sounds like a lot of work - but is probably necessary to demonstrate that there's more to this than qualitative hair-splitting.

This opens a different can of worms: what to do with old holotypes erected during a time of different standards? Even going back to establishing *Callophoca obscura* as a nomen dubium – do we do the same for many cetaceans? *Zarhachis* was stabilized by Kellogg – the type specimen is a vertebra – but the taxonomy is now considered stable around a skull. Do we get rid of *Zarhachis*? *Delphinodon dividum* is not the same morphotype as the type species, *Delphinodon mento*, likely a *Hadrodelphis*-like dolphin; what do we do there? What about *Zygorhiza kochii* – the holotype is pretty scrappy, and Uhen proposed designating a neotype – which Gingerich (2015) railed against for various reasons, and ultimately the ICZN declined to allow neotype designation. I'm not certain which philosophy I agree with, though I have generally opted to follow Kellogg's lead and stabilize old names with the referral of better specimens. I suggest touching up on Romer (1968: Notes and Comments on Vertebrate Paleontology – chapter 1) and mulling it over a bit.

Sorry to be a bit of a stick in the mud on this issue.

Cheers, Robert Boessenecker, Ph.D.

We thank Dr. Boessenecker for his comments, and appreciate the continued discussion around *Mesotaria*. However, we are not convinced by Dr. Boessenecker's rebuttal to our response on this issue. As we will outline below, there is no basis for assigning western North Atlantic material to *Mesotaria* (and no history of use of the name), there is little support

for humeri being taxonomically informative, and the argument (based around intraspecific variation) for referral to *Mesotaria* is circular, as it can also be used as a case against referral to *Mesotaria*.

First, there is no history of referral of the already published skull (USNM PAL 475486) to *Mesotaria ambigua*, and indeed no history of any western North Atlantic fossils to that binomial. After Van Beneden's initial work (1877), for the next 99 years the name *Mesotaria* is only ever used in either faunal or palaeontological lists. It is not until Ray's (1976) paper on pinniped biogeography that *Mesotaria* is mentioned in a taxonomic sense. In this paper, the species *Mesotaria ambigua* was reassigned to the genus *Callophoca* (*Callophoca ambigua*); the basis for this reassignment was not stated. However, no western North Atlantic material was referred to this revised binomial. After that, *Mesotaria* is next revised in Koretsky and Ray 2008. In this paper, Koretsky and Ray formalised the names *Callophoca obscura* and *Mesotaria ambigua* by assigned lectotype specimens, and then subsequently synonymize the name *Mesotaria ambigua* with *Callophoca obscura*. As a result, the U.S. specimens were referred to the name *Callophoca obscura*, not *Mesotaria ambigua*. It is also worth noting that Koretsky and Ray do not outline a diagnosis or give a description of *Mesotaria ambigua*, they only state that *Mesotaria ambigua* represents a male morph of *Callophoca obscura*. Because of this, by declaring *Callophoca obscura* a *nomen dubium* (a move fully supported by Dr. Boessenecker) we are starting with a clean slate for the U.S. material.

Considering this, we should to be very careful when trying to refer more complete material to genera or species based on isolated humeri, especially when the less complete specimens/names are from a different time (late Miocene) and place (the eastern North Atlantic). For example, referring the U.S. specimens (USNM PAL 475486, USNM PAL 534034, USNM PAL 181601) to even just the genus *Mesotaria* has biological consequences, implying relationships and biogeographic patterns. As such, it is critical to refer material to meaningful biological entities.

The next thing to be considered is: are we able to meaningfully compare the *Mesotaria* lectotype to the humerus of USNM PAL 534034? An existing morphometric analysis from Churchill and Uhen (2019) strongly suggests "no". As outlined above in the response to the editor, their morphometric analysis, and their qualitative review of morphological characters of the humerus, demonstrated that humeri are taxonomically uninformative for phocids. This means they have no taxonomic utility as type specimens, and hence shouldn't be used to assign taxa whether via lumping or splitting. Dr. Boessenecker has highlighted that humeri of other pinnipeds can display lots of intraspecific variation, but if this is the case then there is even more concern to refer material on the basis of humeri. If this logic is applied for other existing fossil phocid species, such as *Piscophoca*, *Acrophoca*, or *Pliophoca*, then we might also refer these taxa to *Mesotaria* (despite the fact that their skulls demonstrate they are clearly separate species). Another issue Dr. Boessenecker brings up is sexual dimorphism. Sexual dimorphism is very rare in extant phocids (*Mirounga*, *Cystophora*, *Halichoerus*), and previous studies have demonstrated it is unlikely to have existed in extinct species (Cullen *et al* 2014); as a result, there is minimal *a priori* reasoning to assume that it would be present here, and therefore it should not be considered as a basis for referral.

Attempting to refer the U.S. specimens to *Mesotaria* brings up another issue: what is the official definition of *Mesotaria* in the literature, beyond the lectotype? Neither Van Beneden (1877), Ray (1976), or Koretsky and Ray (2008) outlined a diagnosis for *Mesotaria ambigua*. The only description of the lectotype humerus (Van Beneden 1877) was brief, focused on the proximal end, and mostly compared it to otariids. Therefore, it is entirely unclear what

defines the binomial. As a result, any differences, however slight, should be cause to doubt a referral.

Dr. Boessenecker has also provided some philosophical discussion regarding the treatment of old holotypes. However, the cetacean examples provided are not really relevant to *Mesotaria* as there is little to no history of use of the name, whereas the cetacean types had a long history of use before stabilisation. And *Callophoca* is certainly not stabilised either, with multiple studies (Berta *et al.* 2015, Boessenecker and Churchill 2016, Dewaele *et al.* 2018, Churchill and Uhen 2019) expressing doubt on its treatment. For these reasons, on top of the numerous reasons to doubt referrals to isolated humerus type specimens, we believe that stabilising old names is not the appropriate course to take.

In light of the above points we have raised, we choose to erect a separate species based on the more complete material. Treating it as a new species eliminates any concerns of historical taxonomic ambiguity, and is also scientifically sound as it is not based off humeri. This new species (*Sarcodectes magnus*) has several unambiguous synapomorphies from the skull, as outlined in the diagnosis, and therefore meets the requirements of both RSOS and ICZN for naming a new species.

Because of the above comments and discussion we have also made a few alterations to the manuscript, to ensure that these clarifying points are not restricted to the discussion with the reviewers. The “Comments” subheading under the Systematic Palaeontology section has been expanded upon to include some of the above points we have raised [lines 239-248]. We have also included some additions to the “Implications for monachine taxonomy” discussion [lines 603, 611, 613-617, 623-625, 629-631] to further add some of the above points. Also, in light of the issue of intraspecific variation, and the points we have made above, we have altered table 5 so that we refer the previous *Callophoca* specimens to “*Monachinae* indet”. We realise this slightly changes the response we made last time, but in light of the above discussion it is the conservative conclusion to make. We have also made some slight adjustments to Figure 14 so the labelling is clearer. None of these additions we have outlined above change the conclusions of the paper, and nor do they drastically alter the discussion presented in the paper.